# Tailoring a local acid-like microenvironment for efficient neutral hydrogen evolution

Xiaozhong Zheng[1], Xiaoyun Shi[1], Honghui Ning[1], Rui Yang[1], Bing Lu[1], Qian Luo[1], Shanjun Mao [1], Lingling Xi[1] & Yong Wang [1,2] ✉

Electrochemical hydrogen evolution reaction in neutral media is listed as the most difficult challenges of energy catalysis due to the sluggish kinetics. Herein, the Ir-$H_xWO_3$ catalyst is readily synthesized and exhibits enhanced performance for neutral hydrogen evolution reaction. $H_xWO_3$ support is functioned as proton sponge to create a local acid-like microenvironment around Ir metal sites by spontaneous injection of protons to $WO_3$, as evidenced by spectroscopy and electrochemical analysis. Rationalize revitalized lattice-hydrogen species located in the interface are coupled with $H_{ad}$ atoms on metallic Ir surfaces via thermodynamically favorable Volmer-Tafel steps, and thereby a fast kinetics. Elaborated Ir-$H_xWO_3$ demonstrates acid-like activity with a low overpotential of 20 mV at 10 mA cm$^{-2}$ and low Tafel slope of 28 mV dec$^{-1}$, which are even comparable to those in acidic environment. The concept exemplified in this work offer the possibilities for tailoring local reaction microenvironment to regulate catalytic activity and pathway.

Sustainable electrocatalytic hydrogen evolution reaction (HER) using renewables powered, low-temperature water electrolyzers is promising for the deployment of the hydrogen economy for sustainable energy storage, transportation, and chemical production[1–7]. It is well-established that this reaction starts with the Volmer step, which generates adsorbed hydrogen intermediates ($H_{ad}$) via electrochemical reduction of either hydronium ion (in an acidic medium) or water (in a neutral or alkaline medium). Subsequently, molecular hydrogen is produced by either a Tafel recombination step ($H_{ad} + H_{ad} \rightarrow * + H_2$) or a charge-transfer Heyrovsky step ($H_{ad} + H_2O + e^- \rightarrow * + H_2 + OH^-$)[8–13]. The kinetics of HER in neutral/alkaline medium is much sluggish than that in acidic environment because of the low concentration of protons. Consequently, even the state-of-the-art Pt-based catalyst, shows two to three orders of magnitude lower activity in neutral/alkaline media as compared to acidic media[14]. The kinetics of HER is strongly relevant to both the nature of electrode materials and the local reaction microenvironment in the vicinity of the catalytic sites at electrolyte–solid interface[15–20]. To date, substantial progress has been made in promoting the kinetics of the electrode reactions by ameliorating the catalyst materials through various ways, such as introducing oxophilic active elements[21,22], heterostructure modulating[23,24], strain engineering[25,26], nanoscopic confinement[27,28]. Apparently, in most cases, these conventional strategies can only modulate their electronic states, adsorption capability of intermediates, and thereby catalytic properties in a gradual or mild way. However, these still cannot get rid of the pH-dependent kinetics of HER, leading to the inability to achieve big breakthroughs in the non-acidic electrolyte. Therefore, selecting a suitable system to create a local acid-like environment through multiple physicochemical effects to the maximum extent possible, will provide an alternative way to promote the electrocatalytic performance and guide the higher efficiency electrocatalyst design in non-acidic electrolyte, especially in more challenging neutral media.

As the prototypical example, the reversible and rapid hydrogen doping of $WO_3$ to form $H_xWO_3$ bronze was well reported via electro-chemical electron–proton co-doping[29–33], in which H atoms are incorporated as W-OH species with Brønsted acidity and reducing some $W^{6+}$ to $W^{5+}$ along with charge rearrangement. Protonated $H_xWO_3$ could act as a proton sponge and electron reservoir to create an acid-like microenvironment in the electric double layer, thereby further affecting the reaction barriers and pathway[34–36]. However, the strong

[1]Advanced Materials and Catalysis Group, Center of Chemistry for Frontier Technologies, State Key Laboratory of Clean Energy Utilization, Institute of Catalysis, Department of Chemistry, Zhejiang University, 310028 Hangzhou, P. R. China. [2]College of Chemistry and Molecular Engineering, Zhengzhou University, 450001 Zhengzhou, China. ✉e-mail: chemwy@zju.edu.cn

adsorption of hydrogen by basic oxygen centers and surface frustrated H−H coupling process severely hinder the local acidic species being rationally utilized. Enhancing electrochemical activation capability of $H_xWO_3$ via the addition of cocatalysts is of interest[37–40]. Up to now, it still encounters many problems and challenges, for example, (1) the degree of local acidification of $H_xWO_3$ in neutral media has not been accurately quantified; (2) the synergistic catalysis between local acidic species and co-catalysts has not been fully understood; (3) the enhanced activity cannot be simply attributed to a single optimized catalytic site, and the origin of the activity deserves further investigation. In this work, typical tungsten oxide nanorods-arrays aligned on carbon fibers were deliberately selected as a host material. The $WO_3$ support experienced hydrogen insertion and Ir nanoparticles (NPs) electrodeposition processes to form hybridized Ir−$H_xWO_3$. Experimental results and theoretical calculations deciphered that a high density of surface revitalized WO−H species enrich local proton concentration around Ir sites, and act as scavengers for $H_{ad}$ on the metallic Ir to realize expedited hydrogen evolution rate, which is reflected in the fact that Ir−$H_xWO_3$ exhibits acid-like HER properties in neutral media.

## Results and discussion

### Hydrogen intercalation and storage of $WO_3$

The formation mechanism of $H_xWO_3$ bronze is dependent on the double injection of electrons and protons to the $WO_3$ nanorods grown on carbon fiber paper under cyclic voltammetry measurement (between 0 and −0.35 V vs. reversible hydrogen electrode, RHE) in 1.0 M PBS, as shown in Fig. 1a and Supplementary Fig. 1. Obviously, the color of the $WO_3$ electrode turns from yellow to dark blue (Fig. 1b and Supplementary Movie 1), which is a typical electrochromic process. Correspondingly, UV−vis diffuse reflectance spectra exhibit a prominent increase of absorption at a wavelength above 450 nm after

electrochemical hydrogenation (Fig. 1c), typically resulting from light absorption of the free electrons at oxygen vacancies ($O_v$) and the d-d transition of $W^{5+}$−OH (refs. 31,41). With increasing negative potentials, the electrons accumulated at the electrode/electrolyte interface, thus attracting the protons to reduce the lattice oxygen to produce $O_v$ via the following reaction equation[31], as reflected by a significant electron paramagnetic resonance (EPR) signal at $g = 2.003$ in Fig. 1d.

$$W^{6+}O_3 + 2e^- + H^+ \rightarrow W^{5+}O_2 - O_V + OH^- \qquad (1)$$

Surface active $O_v$ is conductive to adsorbing and subsequently inducing dissociation of water molecules via transferring a proton to adjacent oxygen, thus forming terminal and bridging hydroxyls groups ($W^{5+}$-−OH species) via the following reaction equation:

$$W^{5+}O_2 - O_V + H_2O \rightarrow W^{5+}O_2H - OH \qquad (2)$$

Undoubtedly, ¹H solid-state nuclear magnetic resonance (NMR) spectroscopy (Fig. 1e) and X-ray photoelectron spectroscopy (XPS) results (Supplementary Fig. 2) confirm the above statements. Hydrogen intercalation fundamentally changes the interface properties of $WO_3$. Electrochemical impedance spectroscopy (EIS, Fig. 1f) and Mott−Schottky measurement (M−S, Supplementary Fig. 3) are highly sensitive to detect the interface electron transfer and carrier density[42,43], respectively. An interface transfer resistance and corresponding $dC^{-2}/dV$ significantly decreased after electrochemical protonation, which is ascribed to the high conductivity of $W^{5+}$−OH polaron states and high mobility of hydrogen on the $WO_3$ surface. Furthermore, protonation-driven semiconductor−metal conversion of $WO_3$ arises from a shift in electronic structure that is describable as a repositioning of the W d-bands in Fig. 1g.

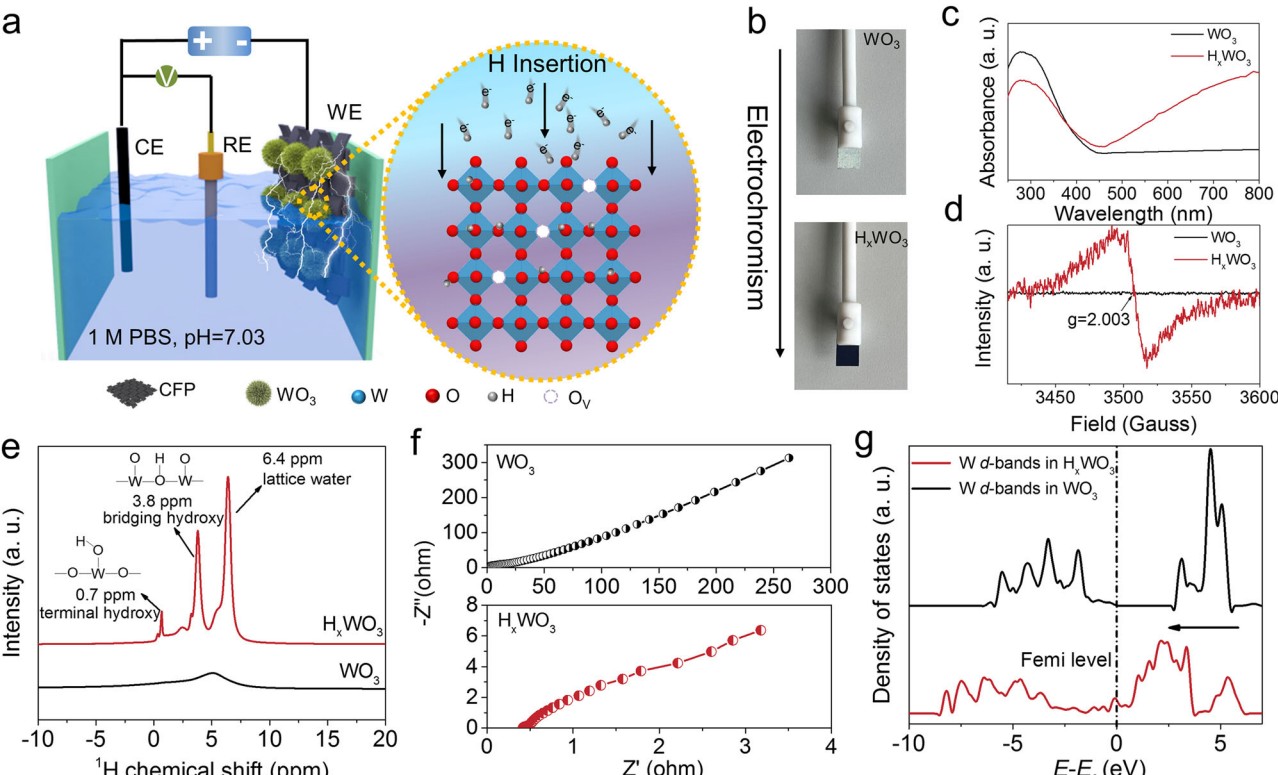

**Fig. 1 | Hydrogen intercalation of $WO_3$. a** Schematic illustration of the hydrogen intercalation process of $WO_3$ to form $H_xWO_3$. **b** Digital pictures of the $WO_3$ working electrode before and after electrochemical hydrogenation. **c** UV−visible diffuse reflectance spectra of $WO_3$ and $H_xWO_3$. **d** EPR signal of oxygen vacancies. **e** ¹H NMR spectra of $WO_3$ and $H_xWO_3$. **f** EIS Nyquist plots of different electrocatalysts measured at open circuit potential from 10 kHz to 0.01 Hz. **g** The pDOS curves of W d-bands of $WO_3$ and $H_xWO_3$.

## Local pH measurements and catalytic behaviors of $H_xWO_3$

We used the rotating ring-disk electrode (RRDE) technique[19,44,45] to quantitatively detect local pH on the $H_xWO_3$ cathode surfaces at different applied potentials in neutral 1.0 M PBS solution (bulk pH is 7.03), along with classical carbon support for comparison (Supplementary Fig. 4–6, details see Methods). On the basis of RRDE detecting method, we found that the pH of the $H_xWO_3$ surface varies from 6.27 to 3.53 as potential decreases from 0.1 to −0.4 $V_{RHE}$, which undoubtedly confirms that $H_xWO_3$ acts as proton sponge to form a local acid-like microenvironment on the electrode surface (Fig. 2a, b). In sharp contrast, the pH of carbon cathode surface is maintained at ~7 in the range of 0.1 to −0.4 $V_{RHE}$ (Fig. 2b). As the potential continued to shift negatively, the pH of $H_xWO_3$ and carbon support surfaces gradually increases and approaches 8.32 and 8.0 at −0.7 $V_{RHE}$, respectively, due to the consumption of the local hydrogen species. Astonishingly, when the bias (−0.7 $V_{RHE}$) is removed, the surface of $H_xWO_3$ and carbon cathodes turn back to steady acidic (pH = 3.73) and neutral (pH = 7.12) states, respectively (Fig. 2c). We then move on to investigate the catalytic behaviors of $H_xWO_3$ in neutral media. Counterintuitively, despite the construction of a unique proton-rich microenvironment, the $H_xWO_3$ grown on CFP material still provides a high overpotential of 548 mV at 10 mA cm$^{-2}$ (Fig. 2d). Following the insertion of hydrogen atoms, H-H coupling must occur to form molecular $H_2$. This can take place through surface-mediated (Tafel step) or water-mediated (Heyrovsky step) routes[8]. In the surface-mediated mechanism, it is hard for two surface hydrogen atoms of $H_xWO_3$ to proceed H-H coupling because active hydrides are separated by relatively long distances (Supplementary Fig. 7), which is further reflected by high Tafel barrier (0.82 eV in Fig. 2f). In the water-mediated mechanism, surface hydrogen atoms form $H_2$ by reacting with water of the electrolyte. The analysis of the Tafel slope for $H_xWO_3$ (Fig. 1e) reveals that it obeys the water-mediated mechanism, but requires a high overpotential (>500 mV) to drive it. As shown by density functional theory (DFT) calculations in Fig. 2f and Supplementary Fig. 8, the poor catalytic activity of $H_xWO_3$ catalyst is

well explained by the fact that surface water-mediated H-H transfer predominates in the early stage of HER and water-mediated H-H coupling occurs only when the overpotential is largely increased. The weak electrochemical activation capability of $H_xWO_3$ severely prevents its hydrides from being properly utilized[46]. To tackle this problem, the construction of $H_xWO_3$ with metal-based materials is considered as a promising design strategy in the following two aspects: (1) interfacial polarization electric field generated by metal and $H_xWO_3$ to activate lattice-hydrogen species; (2) providing additional active sites to promote interfacial synergistic effects between them.

## Synthesis and characterization of catalysts

Based on the above assumptions, Ir–$H_xWO_3$ hybrid electrocatalyst was intentionally manufactured similar to that of $H_xWO_3$ except for adding 0.2 g L$^{-1}$ $IrCl_3$ to 1.0 M PBS solution. $Ir^{3+}$ precursors are progressively reduced to metallic Ir nanoparticles (NPs) under applied negative bias, as revealed by the HER activity enhancement increases with cycles, eventually leveling off after 500 cycles in Supplementary Fig. 9. Ir content in Ir–$H_xWO_3$ hybrid electrocatalyst is 2.8 wt% (or 47 µg cm$^{-2}$ when normalized to geometric area of electrode) determined by inductively coupled plasma-optical emission spectrometry (ICP-OES, Supplementary Table 1). X-ray diffraction (XRD) patterns in Fig. 3a disclose that as-prepared $WO_3$, $H_xWO_3$, and Ir–$H_xWO_3$ electrocatalysts exhibit clear diffraction peaks of hexagonal $WO_3$ (JCPDS No. 85-2460), but no diffraction peaks related to $H_xWO_3$. Notably, the characteristic diffraction peaks of $WO_3$ in $H_xWO_3$ and Ir–$H_xWO_3$ catalysts are shifted toward a low-angle direction (Supplementary Fig. 10), probably ascribing to oxygen defect-induced lattice extension[47]. Meanwhile, for Ir–$H_xWO_3$, the signals of Ir NPs are not detected, possibly due to their small sizes and low content. As revealed in Supplementary Fig. 11, 12 and Fig. 13a–c, after electrochemical protonation, the scanning electron microscopy (SEM) and transmission electron microscopy (TEM) images display that $H_xWO_3$ and Ir–$H_xWO_3$ samples inherit the original $WO_3$ nanorods-like morphology. Comparatively, the TEM images

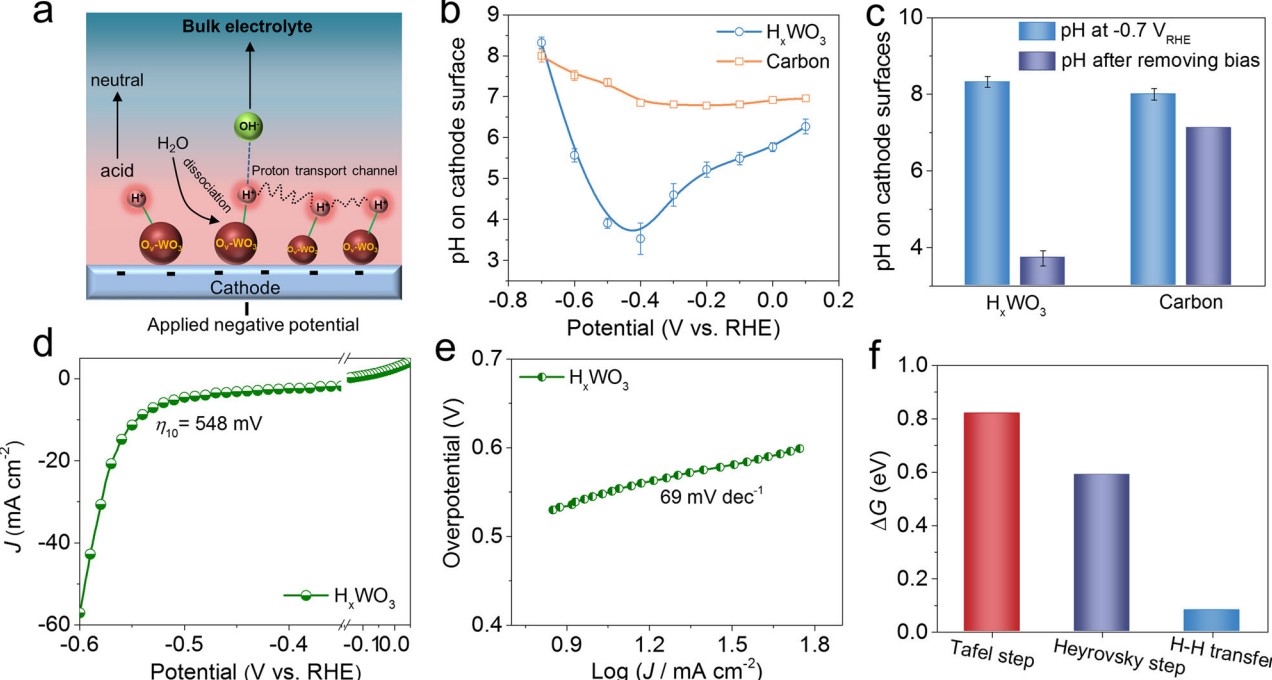

**Fig. 2 | Local pH measurements and catalytic behaviors of $H_xWO_3$. a** Schematic diagram of local acid-like microenvironment generation on $H_xWO_3$ cathode. **b** Measured pH values on $H_xWO_3$ and carbon cathode surfaces at different potentials, respectively. **c** The pH on $H_xWO_3$ and carbon cathode surfaces before and after removing bias (−0.7 $V_{RHE}$). **d** HER polarization curves of $H_xWO_3$ grown on CFP with 95% iR compensation and (**e**) corresponding Tafel plots. **f** Free energy barriers of H-H coupling and H-H transfer in $H_xWO_3$. Note: error bars represent the standard deviation of three independent measurements.

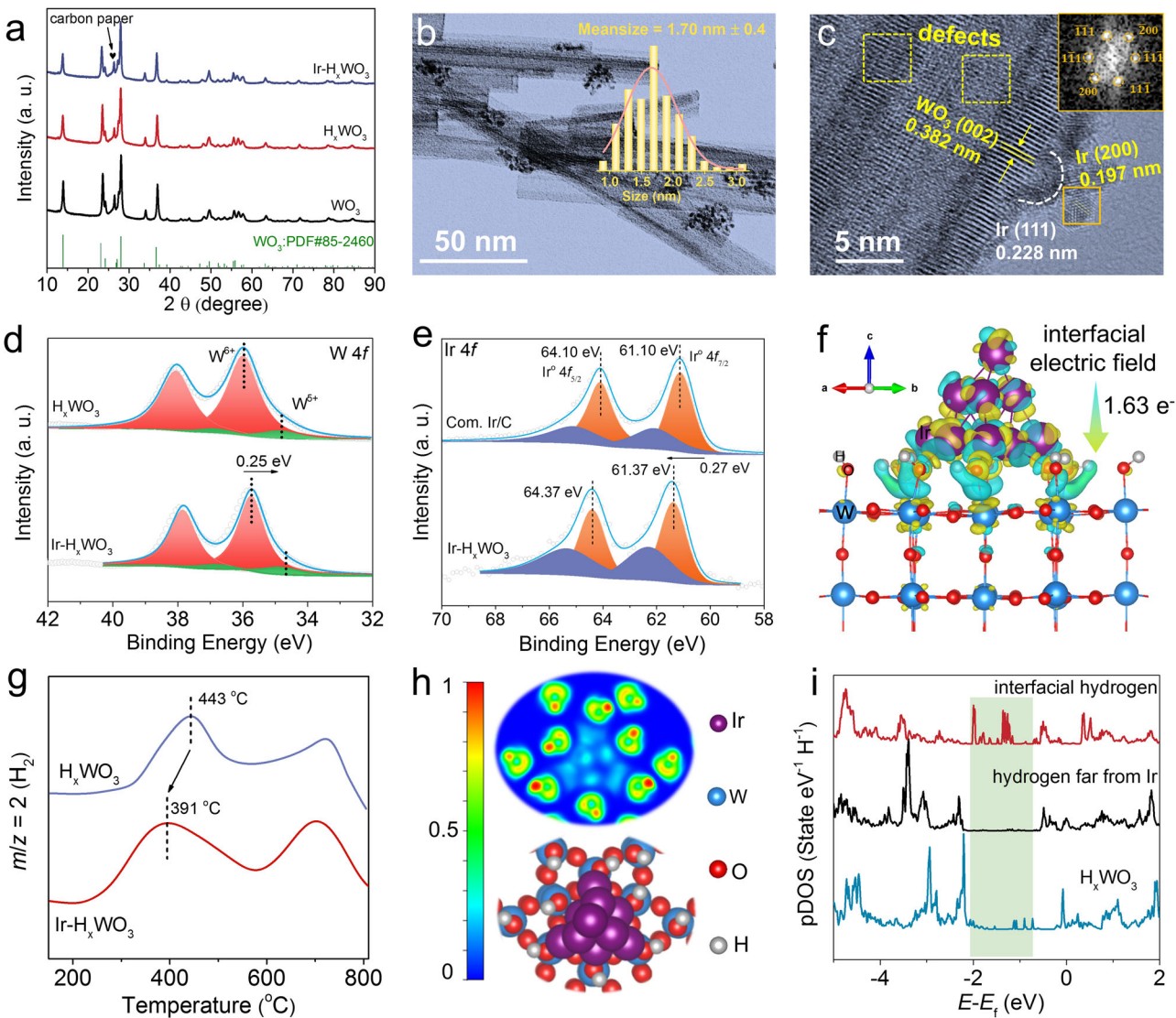

**Fig. 3 | Structural characteristics of catalysts. a** XRD patterns of as-obtained $WO_3$, $H_xWO_3$ and Ir–$H_xWO_3$. **b** Typical TEM image of Ir–$H_xWO_3$, the inert in (**b**) shows the Ir NPs size distribution pattern. **c** HAADF-STEM images of Ir–$H_xWO_3$, with the inset in (**c**) giving the fast Fourier transform of the corresponding Ir NP. **d** High-resolution W $4f$ spectra of Ir–$H_xWO_3$ and $H_xWO_3$. **e** High-resolution Ir $4f$ spectra of Ir–$H_xWO_3$ and commercial Ir/C. **f** The side views of charge density difference in the interface of $Ir_{10}$ clusters supported on $H_xWO_3$. The cyan region reflects an electron-deficient state while the yellow region reflects an electron-rich area, with an iso-value of 0.004 $e^-/Å^3$. **g** Representative TPD-MS thermal desorption profiles for Ir–$H_xWO_3$ and $H_xWO_3$ and the main desorbed specie detected is $H_2$ with $m/e = 2$. **h** Electron localization function evaluations of $Ir_{10}$-$H_xWO_3$. **i** The pDOS curves of 1$s$ orbitals of different hydrogen atoms in $Ir_{10}$–$H_xWO_3$ and pure $H_xWO_3$.

(Fig. 3b and Supplementary Fig. 13d–f) of Ir-$H_xWO_3$ demonstrate that Ir NPs with mean size of ~1.7 nm are embedded on $H_xWO_3$ nanorods. Corresponding EDS elemental mapping images and line-scan profiles (Supplementary Fig. 13g–k) further verify the successful loading of Ir. The Ir NPs exhibit clear lattice fringes with interplanar distances of 1.97 and 2.28 Å, corresponding to the [200] and [111] crystal planes of the [011] face-centered cubic Ir, respectively, further evidenced by the fast Fourier transform (FFT) pattern (Fig. 3c). Meanwhile, the disordered structures identified by deformed and blurred lattice fringes belonging to [002] plane of $WO_3$ are found in Ir-$H_xWO_3$ (Fig. 3c), which is ascribed to a good deal of surface defects induced by hydrogen intercalation during electrochemical activation[48]. Subsequently, XPS was carried out to further analyze the atomic structure and surface valence of catalysts. The typical W $4f$ spectra of $H_xWO_3$ and Ir-$H_xWO_3$ indicate the coexistence of $W^{6+}$ and $W^{5+}$ (Fig. 3d)[30]. Compared with those of $H_xWO_3$, W $4f$ binding energies of the Ir-$H_xWO_3$ show a significant redshift of 0.24 eV. The O 1$s$ split-peak fitting results (Supplementary Fig. 14) reveal that lattice oxygen, surface hydroxyl, and adsorbed water species are detected on the surface of $H_xWO_3$ and Ir-$H_xWO_3$ samples. As excepted, after Ir NPs loading, the O 1$s$ spectrum of Ir-$H_xWO_3$ moves toward lower binding energies. Figure 3e demonstrates the comparison of Ir $4f$ XPS spectra between commercial Ir/C and Ir-$H_xWO_3$. It is noticeable that the binding energy of Ir for Ir-$H_xWO_3$ is positively shifted by 0.27 eV up to 61.37 ($Ir^o$ $4f_{7/2}$) and 64.37 eV ($Ir^o$ $4f_{5/2}$). These results manifested the strong electronic interaction between Ir NPs and $H_xWO_3$ support. Further, the charge density difference diagram of the interface between Ir NPs and $H_xWO_3$ (Fig. 3f) reveals a charge transfer of 1.63 $e^-$ from the Ir atom to $H_xWO_3$ through interfacial Ir-O-W bond, confirming the XPS results. In addition, the planar average potential plots along the $Z$-direction (Supplementary Fig. 15) confirm the interfacial built-in electric fields in Ir-$H_xWO_3$.

Undoubtedly, the strong electric field effect at the interface between Ir NPs and $H_xWO_3$ will affect the lattice-hydrogen species in $H_xWO_3$ support. The $^1H$ NMR spectra (Supplementary Fig. 16) confirm the signals of Ir-$H_xWO_3$ broadens and moves toward higher fields (1.1

and 4.0 ppm) compared to those of $H_xWO_3$ (0.7 and 3.8 ppm), which indisputably reveal that the acid strength of Ir−$H_xWO_3$ material is stronger than that of $H_xWO_3$ (ref. 49). Stronger acid strength also means easier detachment of protons. To prove this statement, temperature-programmed desorption (TPD) coupled with mass spectrometry ($m/z = 2$, $H_2$) was performed to investigate the desorption behavior of lattice-hydrogen species over $H_xWO_3$ and Ir−$H_xWO_3$. As demonstrated in Fig. 3g, the TPD profiles of Ir−$H_xWO_3$ and $H_xWO_3$ are characterized by two typical peaks, assigned to the recombination of surface and bulk hydroxyl groups producing $H_2$ (ref. 50), respectively. It is worth noting that the introduction of Ir accelerates the desorption of lattice-hydrogen species in the $H_xWO_3$ support, as revealed in shifting towards low temperature of Ir−$H_xWO_3$ TPD pattern compared to pure $H_xWO_3$. We next considered the chemical reasons for why Ir metal profoundly changes the behavior of hydrogen desorption over $H_xWO_3$ surfaces. The contour map of electron localization function (ELF) for Ir−$H_xWO_3$ is shown in Fig. 3h, projected over the [001] plane to analyze the electron localization and bond polarity characters[51]. The addition of Ir in the $H_xWO_3$ structure leads to the localized electron density redistribution of WO−H species adjacent to Ir NPs (denoted as interfacial hydrogen species) and the formation of more polar O−H bonds. Contrariwise, O−H bonds far from Ir metal exhibit a considerably covalent character. Bader charge analysis was conducted to quantitatively study the charge distribution. In Supplementary Fig. 17, the average Bader charges value of interfacial hydrogen atoms is 0.341 e⁻, which is significantly lower than that of the hydrogen atoms far away from Ir (0.365 e⁻) and pure $H_xWO_3$ (0.361 e⁻). Further extracting the projected density of states (pDOS) curves of different chemical states of lattice-hydrogen species, as shown in Fig. 3i, it is well noted that interfacial-hydrogen species possess a higher electronic state near the Fermi level compared to pure $H_xWO_3$ and hydrogen species far from Ir metal, indicating that metallic Ir revitalizes the lattice-hydrogen species and enabling it possible to participate in the HER process.

## Electrocatalytic HER performances

Encouraged by a local activated acid-like microenvironment created around Ir NPs in Ir−$H_xWO_3$, it is expected to be able to substantially boost HER activity in non-acidic environments. Thus, the electrocatalytic HER activity of Ir−$H_xWO_3$, commercial 20 wt% Pt/C and 10 wt% Ir/C catalysts were tested in a conventional three-electrode electrochemical cell with $H_2$-saturated 1.0 M PBS as electrolyte (Supplementary Fig. 18), along with acidic HER evaluation in 0.5 M $H_2SO_4$ for comparison. The catalytic activity was obtained from iR-compensated linear sweep voltammetry (LSV) curves, and different levels of iR compensation were considered and presented in Supplementary Fig. 19. To avoid potentiostat oscillation and overcorrected results[52], automatically 95% iR compensation was adopted in this electrochemical test. As expected, in Fig. 4a−c, the HER activity and kinetics of Ir−$H_xWO_3$ under neutral conditions are similar to those in acidic media. Specifically, at an overpotential of 150 mV, Ir−$H_xWO_3$ can deliver 256 mA cm⁻² in neutral media, but it only increases to 277 mA cm⁻² under acidic media. The close Tafel slope values (neutral: 28 mV dec⁻¹; acidic: 25 mV dec⁻¹) indicate a similar hydrogen evolution pathway (Tafel step: $H_{ad} + H_{ad} \rightarrow * + H_2$). However, striking performance discrepancy is observed in well-known Pt/C and Ir/C electrocatalysts in different electrolytes. In acidic media, the current density of Pt/C and Ir/C is 2.62 and 6.18 times higher than those in neutral medium at 150 mV overpotential. The significantly reduced Tafel slopes manifests that HER catalytic kinetics is strongly related to the proton concentration. These results corroborate that for conventional carbon-supported metal catalysts exhibit highly pH-dependent catalytic activity, while $H_xWO_3$ plays as proton sponge to create acid-like microenvironment around metal catalysts to obtain thermodynamically favorable catalytic activity and accelerated reaction kinetics in a proton-deficient neutral media. It is noteworthy that the

Ir−$H_xWO_3$ ($\eta_{10}$ = 20 mV; Tafel slope: 28 mV dec⁻¹) is among the best HER electrocatalysts reported in the neutral medium (Fig. 4d and Supplementary Table 2). Accelerated HER kinetic of Ir−$H_xWO_3$ is also reflected by the smaller charge-transfer resistance (the width of the semicircle, 2.5 Ω, Fig. 4e). To further identify its high intrinsic HER activity, being normalized to the per milligram noble-metal loading, Ir−$H_xWO_3$ (5.46 A mg$_{Ir}$⁻¹) exhibits 4.3 and 13.3 times higher mass activity than those of Pt/C (1.27 A mg$_{Pt}$⁻¹) and Ir/C (0.41 A mg$_{Ir}$⁻¹) at an overpotential of 150 mV (Fig. 4f).

High-speed imaging experiments prove that the superhydrophilic structure (surface W−OH group) of Ir−$H_xWO_3$ better suppresses bubble coalescence and enhances bubble release compared to commercial Pt/C counterpart (Fig. 4g and Supplementary Movie 2−3), so Ir−$H_xWO_3$ was proposed to have better catalytic performance for water splitting, especially at high current densities. Additionally, we performed the stability test for fully assessing the catalyst via the accelerating degradation technique (ADT) and potential-constant electrolysis. Supplementary Fig. 20 shows the polarization curve of Ir−$H_xWO_3$ manifests negligible shift after 10,000 ADT cycling tests. The long-term durability testing on the Ir−$H_xWO_3$ catalyst by a static chronopotentiometry test (at 10 mA cm⁻² in Fig. 4h) even represents a relatively stable horizontal line with an overpotential increase of only 14 mV over 100 operating hours, while Pt/C exhibits a clipping activity decay within a few hours, demonstrating the good activity retention of Ir−$H_xWO_3$. However, it is still a challenge to develop efficient catalysts working at large current density for neutral water splitting. Amazingly, Ir−$H_xWO_3$ catalyst maintains operation stability at a large current density of 500 mA cm⁻² over 40 h, outperforming most of the recently reported landmark catalysts (Supplementary Table 2). Both the structure and composition of Ir−$H_xWO_3$ remain unchanged before and after the HER stability test as examined by XRD, SEM, HRTEM and ICP-OES (Supplementary Fig. 21 and Table 2). For practical application, natural seawater splitting performance is also evaluated. As expected, in Supplementary Fig. 22, Ir−$H_xWO_3$ exhibits good HER performance in natural seawater, requiring an overpotential of 150 mV to produce 10 mA cm⁻² of catalytic current density. This is lower than that of Pt/C (295 mV) and other state-of-art catalysts (Supplementary Table 3). In summary, the efficient Ir−$H_xWO_3$ catalyst under neutral/near-neutral conditions is expected to be applied for next-generation water-splitting technologies.

## Exploration of catalytic mechanisms

To gain more insights into the original active sites of electrocatalysts, the operando electrochemical Raman spectra were then recorded to investigate the neutral HER behavior at the Ir−$H_xWO_3$ and $H_xWO_3$ surfaces (Supplementary Fig. 23). For $H_xWO_3$ sample, as indicated in Fig. 5a, b, the intensity of typical peaks at 762 cm⁻¹, 714 cm⁻¹ (stretching vibrations of O−W−O skeleton) and 926 cm⁻¹ (stretching vibrations of terminal W=O bond) sharply weaken from open circuit potential (OCP) to −0.1 V$_{RHE}$. Further decreasing potential (−0.15 to −0.3 V$_{RHE}$), all of these characteristic peaks disappear, accompanied by the increasing intensity of the ~1580 and 2715 cm⁻¹ band from the bending and stretching vibration modes of WO−H species[53–55]. This observation confirms that hydrogen insertion occurs in $WO_3$ in response to the cathodic voltage, where the H atoms are incorporated into the Brønsted acidic W⁵⁺−OH groups, and the hydrogen storage saturates after the potential reaches −0.15 V$_{RHE}$. However, the lack of H−H coupling sites and weak electrochemical activation prevent the surface hydride from being effective as $H_2$ molecular from the surface, which is reflected by its poor catalytic activity. Additionally, the variation trends of the two broad bands at 3249 and 3407 cm⁻¹ (Fig. 5a, c), assignable to tetrahedrally and trihedrally coordinated water at the interface[56,57], suggest that the process of hydrogen incorporation is closely related to the activation of interfacial water. In stark contrast, with regard to Ir-modified $H_xWO_3$ electrocatalyst (Fig. 5d, e), the Raman signals of

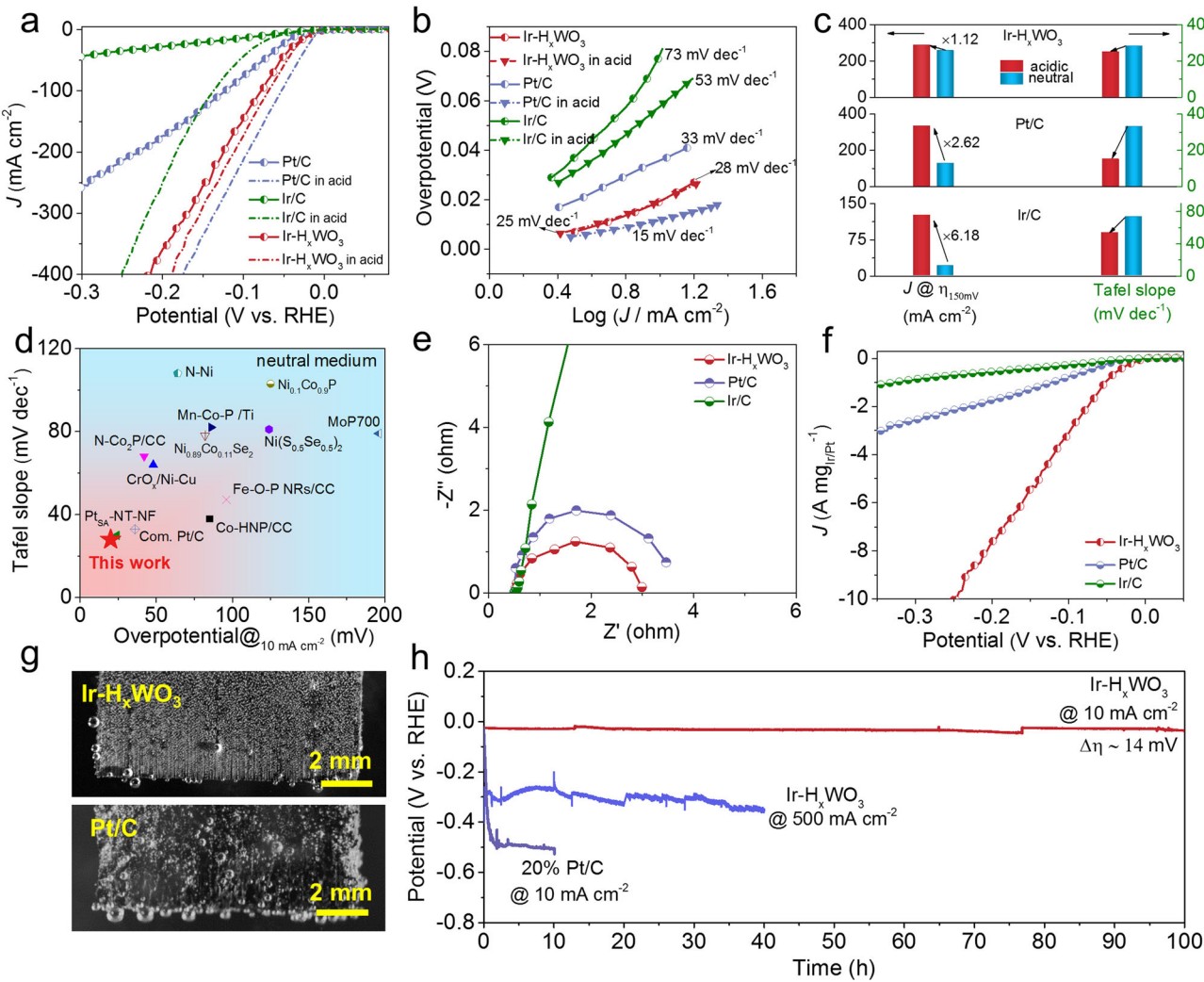

**Fig. 4 | Electrocatalytic HER performances. a** HER polarization curves of different electrocatalysts in 1.0 M PBS and 0.5 M $H_2SO_4$ at 2 mV s$^{-1}$ scan rate. **b** Tafel plots obtained from the polarization curves. **c** Comparison of current density (@$\eta_{150\,mV}$) and Tafel slopes of different electrocatalysts in acidic and neutral media. **d** Comparison of neutral HER activity between Ir–H$_x$WO$_3$ and recently reported electrocatalysts. **e** EIS Nyquist plots of different electrocatalysts measured at −0.03 $V_{RHE}$ from 10 kHz to 0.01 Hz in 1.0 M PBS. **f** Mass activity of electrocatalysts. **g** The $H_2$ bubbles desorption behaviors of Ir–H$_x$WO$_3$ and commercial Pt/C at 100 mA cm$^{-2}$. **h** Chronopotentiometry tests of Ir–H$_x$WO$_3$ (@10 and 500 mA cm$^{-2}$) and commercial 20% Pt/C (@10 mA cm$^{-2}$) in 1.0 M PBS. All electrochemical data were corrected for 95% iR drop.

W–O–W, W = O and WO–H progressively weaken over the whole potential range (OCP to −0.3 $V_{RHE}$), which further illustrates that the Ir–H$_x$WO$_3$ depleted surface WO–H species during HER process to alleviate deep-hydrogenation of H$_x$WO$_3$ supports. Moreover, compared with H$_x$WO$_3$, the apparent increased intensity of O–H stretching vibrations signals at 3249 and 3407 cm$^{-1}$ and the appearance of H–O–H bending vibration bands at 1650 cm$^{-1}$ are detected[57] (Fig. 5d, f), indicating that Ir NPs enhance the adsorption of interfacial water molecules. As subsequent HER going on, the activity of $H_2O$ continued to decrease, verifying that Ir serves as the site of water dissociation. Based on the results of operando electrochemical Raman spectra, it is plausible that there may be possible synergistic catalysis between Ir and lattice-hydrogen species derived from H$_x$WO$_3$.

To verify this hypothesis, first of all, selectively poisoning experiments were conducted. Based on the well-established Li$^+$-ion-exchange method[35] (WO–H + Li$^+$ → WO–Li + H$^+$), Ir–H$_x$WO$_3$ electrocatalyst was immersed in 0.5 M LiNO$_3$ solution for 12 h to completely realize H$^+$–Li$^+$ exchange. As shown in Fig. 6a, after Li$^+$ poisoning, the striking activity decay of Ir–H$_x$WO$_3$ is observed, highlighting the importance of the local acidified environment for HER activity. The

same process was performed on Pt/C and H$_x$WO$_3$ support in Supplementary Fig. 24 for comparison to confirm Li$^+$ selectively poisons the WO–H species and excludes its effect on the metal sites. In addition, it is well known that metal catalytic sites are very sensitive and reactive to SCN$^-$, which can poison metal-centered catalytic sites by coordinating with metallic species strongly[58]. The HER polarization plots of Ir–H$_x$WO$_3$ show the drastic negative shifts of potentials after the introduction of 20 mM SCN$^-$ ions into the electrolyte (Fig. 6a). For comparison, after SCN$^-$ treatments for Pt/C and H$_x$WO$_3$ support in Supplementary Fig. 25, an obvious activity reduction of Pt/C is detected and has no effect on H$_x$WO$_3$ support. Then, the Tafel slopes of Ir–H$_x$WO$_3$ catalyst before and after Li$^+$ or SCN$^-$ poisoning are compared to analyze the change of the catalytic reaction pathway. As shown in Fig. 6b and Supplementary Fig. 26, after WO–H replaced by WO–Li, the value of Tafel slope of catalyst increases from 28 to 89 mV dec$^{-1}$, manifesting that the rate-determining step (RDS) of the reaction changes from Tafel ($H_{ad} + H_{ad} \rightarrow * + H_2$) to Heyrovsky step ($H_{ad} + H_2O + e^- \rightarrow * + H_2 + OH^-$)[12] and elucidating that the presence of lattice-hydrogen alters the catalytic mechanism. Furthermore, after poisoning the metal sites, the RDS transforms into Volmer step (158 mV dec$^{-1}$,

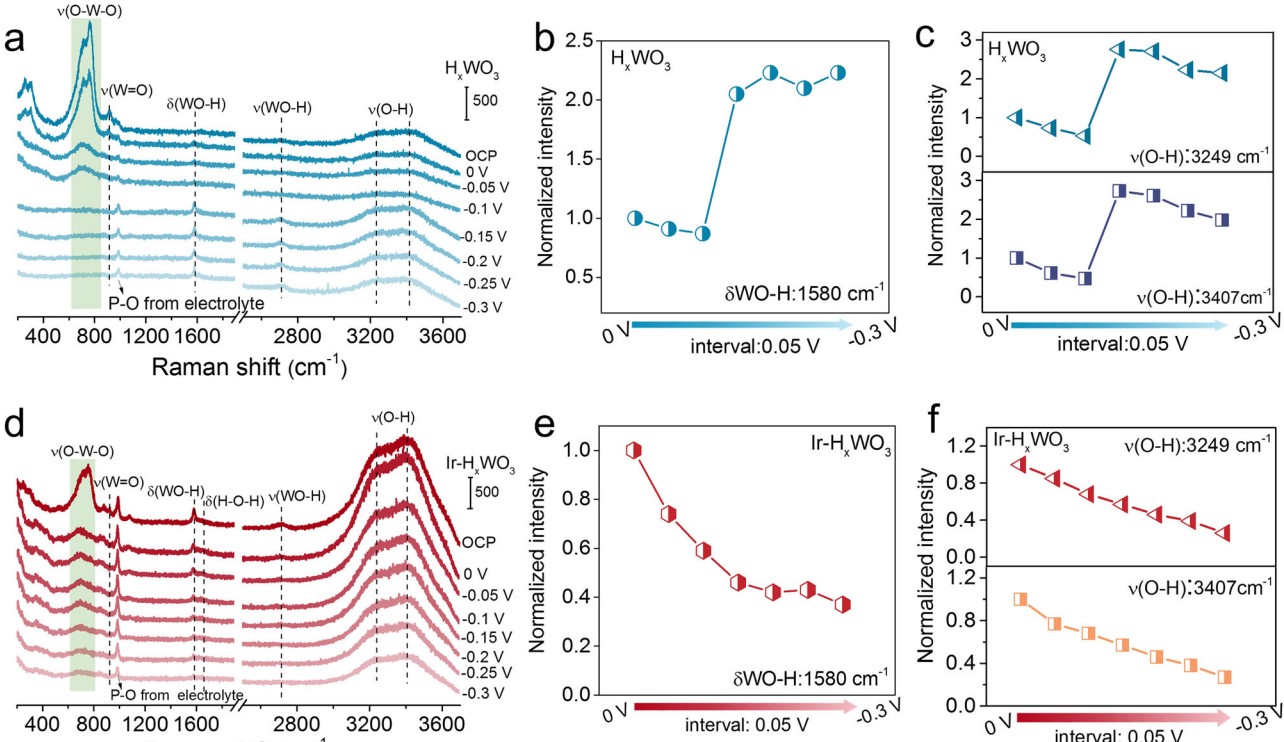

**Fig. 5 | Operando electrochemical Raman measurements. a** Operando electrochemical Raman spectra of $H_xWO_3$ from OCP to −0.3 $V_{RHE}$ and corresponding normalized intensity of WO−H (**b**) and O−H (**c**) signals under different HER potentials. **d** Operando electrochemical Raman spectra of Ir−$H_xWO_3$ from OCP to −0.3 $V_{RHE}$ and corresponding normalized intensity of WO−H (**e**) and O−H (**f**) signals under different HER potentials.

$^*H_2O + e^- \rightarrow ^*H + OH^-)$[12], significantly unveiling that Ir metal site is the response for the water dissociation, which is in line with the results of operando Raman spectra.

Further depicting the rate-limiting step, kinetic isotope effect (KIE) experiments were performed. The LSV curves and corresponding KIE values ($JH_2O/JD_2O$ at −0.1 $V_{RHE}$) are recorded from Ir−$H_xWO_3$ in 1.0 M PBS aqueous electrolyte and 1.0 M PBS $D_2O$ electrolyte (Fig. 6c, d). Interestingly, the LSV plots of Ir−$H_xWO_3$ demonstrate the negative shifts of potentials after several HER cycles and eventually stabilize after 7th LSV scan. These results reveal that the initial KIE effect (2.57) is possibly derived from the O−D bond activation/dissociation of deuterium water, and then surface active WO−H species are gradually replaced by WO−D, resulting in worse hydrogen transfer kinetics[59], which can be explained by the presence of an obvious induction period and corresponding increased KIE value up to 5.16. To further confirm the KIE effect, the pre-synthesized Ir−$D_xWO_3$ (see Methods) also manifests a similar induction period in 1.0 M PBS ($H_2O$) electrolyte, showing enhanced HER activity (Supplementary Fig. 27). The same process was carried out for Pt/C in Supplementary Fig. 28 for comparison. Pt/C only exhibits the typical isotope effect (KIE = 3.79 at −0.1 $V_{RHE}$) and does not have the above-mentioned induction period phenomenon. Compared with Pt/C, the large KIE value variations of Ir−$H_xWO_3$ ($2.57_{initial} < 3.79_{Pt/C} < 5.16_{final}$) indicate the unsubstitutable role of lattice-hydrogen species in the HER process. Combining operando Raman measurements, selective poisoning and kinetic isotope effect experiments, it unambiguously unveiled that a potential neutral HER mechanism involved lattice WO−H species and metallic Ir site synergistic catalysis pathway in Ir−$H_xWO_3$.

Density functional theory (DFT) calculations were then performed to probe the specific reaction pathway of Ir−$H_xWO_3$. Based on the above catalyst characterization, the computational model adopt $Ir_{10}$ cluster supported on $WO_3$ (002) with terminal oxygen saturated with

hydrogen atoms to form experimentally validated WO−H species. The possible HER pathway assisted by support-derived lattice-hydrogen species (WO−H) is systematically investigated in comparison to the traditional HER process (Fig. 7a). The adsorption and dissociation of the $H_2O$ molecule were first examined. The absorbed free energy of $H_2O$ at different Ir sites ($Ir_1$, $Ir_2$, and $Ir_3$ correspond to the interfacial Ir, subinterfacial Ir and bulk Ir, respectively) are calculated (Supplementary Fig. 29). Clearly, $Ir_3$ active site is found to be the most favorable (or dominant) site for $H_2O$ adsorption with more negative $\Delta G_{H2O^*}$ (−0.16 eV) as compared with that of $Ir_1$ (0.29 eV) and $Ir_2$ (0.12 eV). Subsequently, $^*H_2O$ dissociation is exothermic by 0.66 eV, and the formed H atom is adsorbed on the neighboring $Ir_2$ site. The Tafel slope for Ir−$H_xWO_3$ suggests that hydrogen evolution over this material should occur via recombination of two H atoms. Then, two Tafel pathways are considered: (1) Traditional Tafel pathway (green line). Another $H_2O$ is adsorbed on $Ir_3$ and cleaved to $^*H$ and hydroxyl; two H atoms on $Ir_2$ site are preferentially coupled together by Tafel reaction with a high free energy barrier (0.85 eV). (2) Interfacial Tafel pathway involved lattice-hydrogen (violet line). $^*H$ on the $Ir_2$ site undergoes two-steps hydrogen transfer to the $Ir_1$ site with a substantially low energy barriers (0.41 eV), subsequent Tafel step for $Ir_1$−$^*H$ and adjacent WO−$^*H$ species experiences an exothermic process (−0.08 eV), which contributes to fast hydrogen production rate, as obtained experimentally. Moreover, this unique and dominant interfacial dehydrogenation site is further corroborated by $H_2$-TPD experiments in Supplementary Fig. 30.

Based on the conclusions obtained from the above experimental and theoretical studies, we propose a possible interfacial hydrogen-evolution pathway mediated by lattice hydrogen of neutral HER catalyzed by hybridized Ir−$H_xWO_3$ (Fig. 7b). The Ir metal component in the hybridized Ir−$H_xWO_3$ possesses superior electrocatalytic activity toward the Volmer process and is expediently utilized to strongly

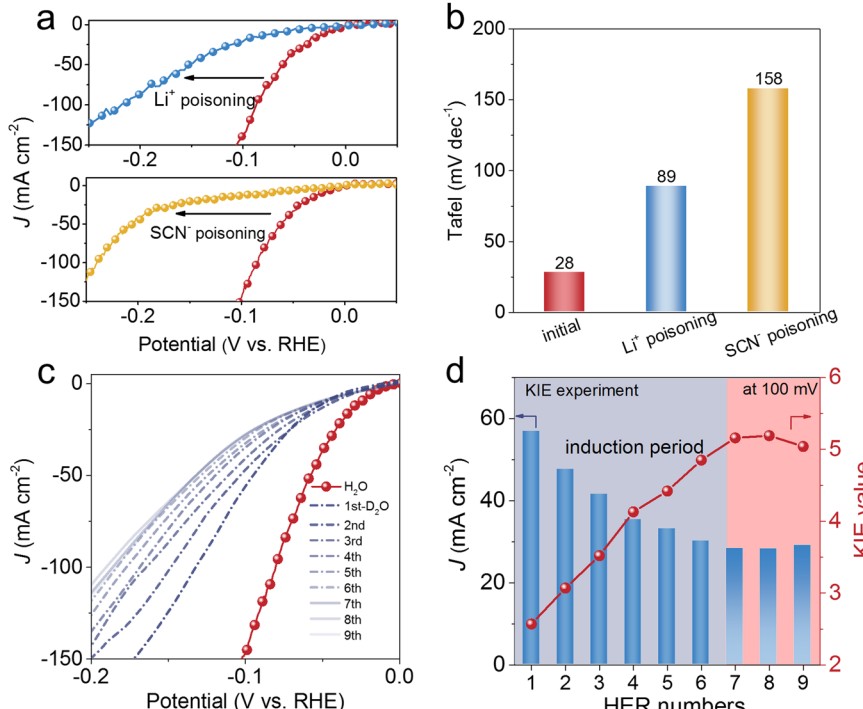

**Fig. 6 | Poisoning and KIE experiments. a** The LSV curves of Ir−$H_xWO_3$ before and after poisoning metallic Ir sites and WO−H sites, respectively. **b** The Tafel slope comparisons of Ir-$H_xWO_3$ before and after poisoning experiments. **c** The LSV curves of Ir-$H_xWO_3$ in 1.0 M PBS ($H_2O$ and $D_2O$) electrolyte. **d** Calculated KIE values ($JH_2O/JD_2O$) under 100 mV overpotential. All electrochemical data were corrected for 95% iR drop.

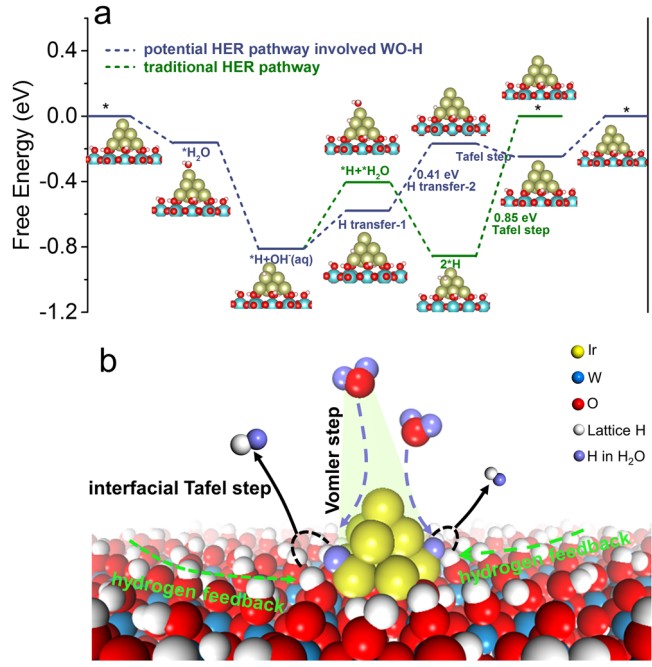

**Fig. 7 | DFT calculations and catalytic mechanisms. a** Free energy profiles for HER over $Ir_{10}$-$H_xWO_3$ via different reaction pathways. **b** Schematic illustrating the proposed reaction mechanism on Ir-$H_xWO_3$.

adsorb $H_2O$, effectively catalyze the dissociation of $H_2O^*$ to generate interfacial Ir-$^*H$. The thermodynamically favorable Tafel process is advantageously utilized to efficiently combine the interfacial Ir-H$^*$ and neighboring reactive WO-H$^*$ into $H_2$. The hydrogen-deficient state at the interface can be eliminated by the hydrogen transfer on the surface

of $H_xWO_3$ to replenish hydrogen, thereby realizing the closed loop of the entire catalytic reaction. Additionally, Supplementary Movies 4 and 5 visually demonstrate the continuous replenishment of hydrogen on Ir-$H_xWO_3$ surface after removing bias, along with Pt/C for comparison. Hence, the exceptional neutral HER electrocatalytic performance of Ir-$H_xWO_3$ stems from the coherent synergism of Ir and in situ formed lattice-hydrogen components. Inspired by the local acid-like microenvironment created by $H_xWO_3$, it is expected to extend "proton sponge" effect of $H_xWO_3$ support to other $M$−$H_xWO_3$ systems ($M$ = Pt, Ru, Pd, Co, Ni). As expected in Supplementary Fig. 31, compared with conventional $M$-Carbon catalysts, the significantly reduced overpotentials and accelerated reaction kinetics on $M$-$H_xWO_3$ systems ($M$ = Ru, Pt, Pd, Co, Ni) confirm that the local acid-like microenvironment provided by $H_xWO_3$ fundamentally enhances the intrinsic HER activity of catalysts in thermodynamically unfavorable neutral media.

## Neutral water electrolysis device performance

To highlight the practical significance of localized acidification environmental engineering for neutral water reduction, we further integrated bifunctional Ir−$H_xWO_3$ catalysts into a membrane electrode assembly (MEA) as cathode and anode materials and assembled an actual anion-exchange-membrane water electrolysis device (Fig. 8a). The current density of the MEA composed of Ir−$H_xWO_3$/CFP($\pm$) is much higher than that of the MEA composed of benchmark commercial (−)Pt/C + Ir/C(+) under the same cell voltage. At a current density of 10 mA cm$^{-2}$, the cell voltage is 1.78 V for Ir−$H_xWO_3$/CFP($\pm$)-based MEA system, which is much less than that of 2.05 V for benchmark commercial (−)Pt/C + Ir/C(+)-based MEA setup (Fig. 8b). Significantly, the Ir−$H_xWO_3$/CFP($\pm$) MEA can be stably operated for at least 40 h at a larger current density of 150 mA cm$^{-2}$ (Fig. 8c), demonstrating unprecedented application prospects.

In summary, we have developed a facile electrochemical method to synthesize a highly unique Ir−$H_xWO_3$ catalyst for efficient water splitting in challenging neutral media. The intentionally created local

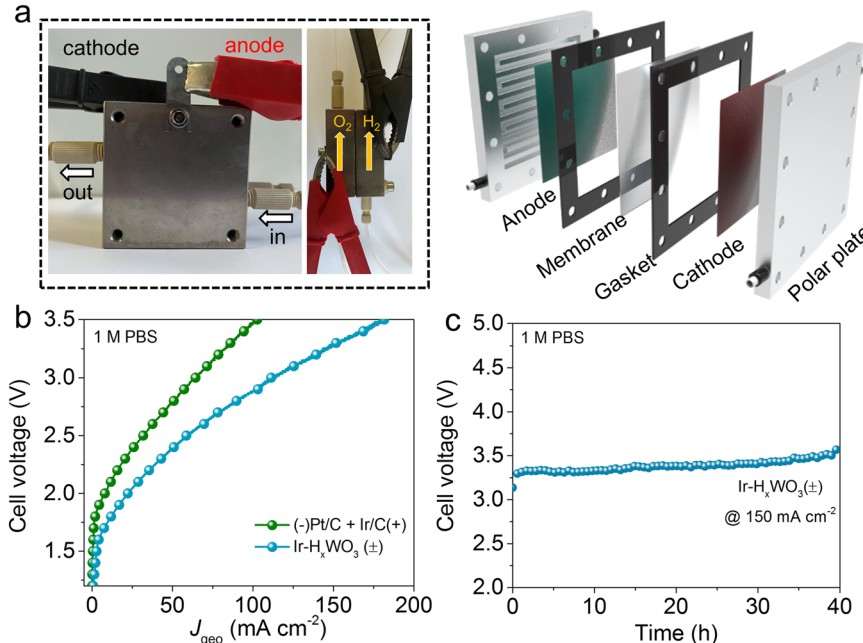

**Fig. 8 | Neutral water electrolysis device performance. a** Photographs and schematic illustration of membrane electrode assembly (MEA) electrochemical reactor, the geometric area of the electrode is 4 cm². **b** Neutral water splitting performance of the commercial (−)Pt/C + Ir/C(+) and Ir−$H_xWO_3$/CFP(±) MEA setups at room temperature without iR compensation. **c** Stability tests of the Ir−$H_xWO_3$/CFP(±) MEA.

acid-like microenvironment around Ir by spontaneous insertion of protons into $WO_3$ was profoundly verified by the electrochromic process and corresponding physicochemical characterizations. Operando Raman measurements, selective poisoning and kinetic isotope effect experiments confirm the coherent synergism between Ir and lattice-hydrogen species of $H_xWO_3$, that is, the Volmer process is drastically boosted at Ir site to form Ir−H*, followed by spontaneous recombination of Ir−H* and neighboring revitalized WO−H* species to form $H_2$ molecular via interfacial Tafel step, as verified by theoretical calculations, thereby keeping the reaction at a high rate. Consequently, the Ir−$H_xWO_3$ catalyst breaks the traditional pH-dependent kinetics limitations compared with conventional Ir/C and Pt/C systems, showing a low overpotential of 20 mV at 10 mA cm⁻² and Tafel slope of 28 mV dec⁻¹ in neutral media, closing to those in acidic media. A neutral water electrolysis device assembled with Ir−$H_xWO_3$(±) realizes a cell voltage of 1.78 V at a current density of 10 mA cm⁻² and a high durability of 40 h at a larger current density of 150 mA cm⁻². Therefore, our study provides insight into tailoring the local reaction environment to design high-performance catalysts in a more rational and precise way.

## Methods
### Chemicals
Iridium(III) chloride hydrate (IrCl₃·xH₂O, 298.58), Ammonium metatungstate ((NH₄)₆H₂W₁₂O₄₀, 2974.32), Ruodium(III) chloride hydrate (RuCl₃·xH₂O, 207.43), Palladium(II) chloride (PdCl₂, 177.32), Chloroplatinic acid hexahydrate (H₂PtCl₆·6H₂O, 517.91), Nickel nitrate hexahydrate (Ni(NO₃)₂·6H₂O, 290.78), Cobalt nitrate hexahydrate (Co(NO₃)₂·6H₂O, 291.03) and Citric acid (C₆H₈O₇, 192.13) were purchased from Aladdin Inc. Commercial 20 wt% Pt/C, 10 wt% Ir/C, 5 wt% Ru/C and 5 wt% Pd/C were purchased from Sigma-Aldrich. The commercial carbon fiber paper (CFP) was purchased from Shanghai Hesen Electric Co.

### Synthesis of $WO_3$
Typically, the $WO_3$ grown on CFP was papered by traditional hydrothermal processes followed by heating treatment. The details are as follows: 0.89 g ammonium metatungstate and 0.18 g citric acid were dissolved in 35 mL deionized water and stirred to form a clear solution. A piece of carbon fiber paper (CFP, approximately 2 cm × 4 cm × 0.2 mm) was carefully cleaned with concentrated HNO₃ solution in an ultrasound bath for several minutes. Then, the CFP was cleaned successively by deionized water, acetone, and absolute ethanol. After the cleaning, the CFP was dried at 60 °C for 30 min. Then, it and the aqueous reagent solution were placed together in a 50 mL Teflon-lined stainless-steel autoclave, which was sealed and maintained at 180 °C for 12 h. The as-synthesized material was then taken out, ultrasonically cleaned for 2 min in deionized water, and dried under vacuum at 70 °C overnight. Finally, the collected sample was placed in muffle furnace and heated to 500 °C at a heating rate of 5 °C min⁻¹ and held for 4 h to obtain $WO_3$ sample.

### Synthesis of Ir−$H_xWO_3$
Ir−$H_xWO_3$ was synthesized by a potential-cycling method, which was performed using a CHI 760E electrochemical workstation (Shanghai CHI Instruments Company) and a standard three-electrode cell. 1 M PBS (50 mL) contained 0.2 g L⁻¹ iridium chloride was taken as electrolyte solution. $WO_3$ precursor, a graphite rod and a saturated calomel electrode (SCE) were used as the working electrode (WE), counter electrode (CE) and reference electrode (RE), respectively. On the WE and CE, different potential cycles (100, 200, 300, 400, 500, and 1000) can be carried out between −0.35 and 0 V vs RHE at a scan rate of 100 mV s⁻¹. We have found that 500 cycles can convert the precursor sample to Ir−$H_xWO_3$.

### Synthesis of $M$−$H_xWO_3$ ($M$ = Pt, Ru, Pd, Ni, and Co)
$M$−$H_xWO_3$ ($M$ = Ru, Pt, and Pd) were synthesized by the same approach as the preparation of Ir−$H_xWO_3$, except that metal precursor was changed to ruodium chloride, chloroplatinic acid or palladiump chloride, respectively. Commercial 20 wt% Pt/C, 5 wt% Ru/C and 5 wt% Pd/C were used as comparison samples. $M$−$H_xWO_3$ ($M$ = Ni and Co) were prepared as following steps. 400 μL 20 g L⁻¹ nickel nitrate or cobalt nitrate was dropped onto 1 × 1 $WO_3$/CFP, and then dried up. The resulting material was reduced in tube furnace at 500 °C for 3 h in H₂

atmosphere (flow rate: 50 sccm). The same approach was conducted on pure CFP to obtain Ni and Co comparison samples.

## Synthesis of $H_xWO_3$

$H_xWO_3$ was synthesized by the same approach as the preparation of Ir–$H_xWO_3$, except that the pure 1.0 M PBS was taken as electrolyte solution.

## Synthesis of Ir–$D_xWO_3$

Ir–$D_xWO_3$ was synthesized by the same approach as the preparation of Ir–$H_xWO_3$, except that the 1.0 M PBS ($D_2O$) contained 0.2 g $L^{-1}$ iridium chloride was taken as electrolyte solution.

## Characterization

Powder X-ray diffraction (XRD) was performed on a Bruker D8 Advance diffractometer (Cu Kα, λ = 1.5418 Å) at the operating voltage of at 40 kV and current of 40 mA. TEM and SEM images were taken on a Hitachi-7700 microscope and Hitachi S-4800 microscope, respectively. A JEOL JEM-2100F with a 200 kV acceleration voltage was used to characterize the HRTEM and HAADF-STEM. Electron paramagnetic resonance (EPR) measurements were performed on a Bruker A300 EPR spectrometer. The UV–Vis diffuse reflectance spectra of the samples were obtained by the UV-3600 spectrophotometer in the region of 200–800 nm, with the $BaSO_4$ powder as background. $^1H$ solid-state nuclear magnetic resonance (NMR) measurements were performed at room temperature on a Bruker 400 MHz NMR spectrometer. The content of noble metals in the samples was determined by a Thermo IRIS Intrepid II inductively coupled plasma-optical emission spectrometry (ICP-OES). X-ray photoemission spectra (XPS) were obtained on an Escalab 250XI spectrometer with a monochromatized Al Kα source (1486.6 eV) and a base pressure in the lower $2 \times 10^{-7}$ mbar range, and XPS spectra were calibrated with the C 1$s$ peak at 284.8 eV. High-resolution XPS spectra were acquired with an analyzer pass energy of 40 eV. The XPS spectra were fitted after subtraction of a Shirley background with the available XPSPEAK 4.1 software.

## Temperature-programmed desorption-mass spectroscopy (TPD-MS) measurement

In the experiment to investigate the effect of Ir on hydrogen species in $H_xWO_3$ support, 100 mg of catalyst sample with grain sizes of 60–80 mesh was loaded in a quartz reactor and pretreated with a ultra-pure Ar gas (with a flow rate of 30 ml $min^{-1}$) at 200 °C for 2 h to remove the surface-adsorbed species. The pretreated sample was then cooled to room temperature, following which the sample was heated to 800 °C under the Ar gas flow (30 ml $min^{-1}$) at a heating rate of 2 °C $min^{-1}$.

In the experiments to investigate hydrogen desorption sites, after Ar pretreatment, $H_2$ adsorption was carried in $H_2$ gas flow (30 ml $min^-$) for 1 h, followed by purging with Ar gas (30 ml $min^{-1}$) for 1 h to remove the physically adsorbed $H_2$ on the sample surface. Finally, a programmed temperature desorption test under Ar gas flow was performed.

The identification of species desorbed during the thermal process was done by re-analyzing the total ion spectra for the specific mass. The main desorbed specie detected is $H_2$ with $m/e = 2$.

## Electrochemical measurements

All electrochemical measurements were performed on a CHI 760E electrochemical workstation (Shanghai CHI Instruments Company) at room temperature. A graphite rod and a saturated calomel electrode were used as the counter electrode and the reference electrode, respectively. The calibration procedure for the reference electrode (SCE) was presented in Supplementary Fig. 32. Ir–$H_xWO_3$ Samples were cut into 1 × 1 $cm^2$ and directly used as the working electrode. For commercial Pt/C, Ir/C, Ru/C and Pd/C electrodes, 10.0 mg catalyst powder and 25 μl Nafion solution (5 wt%) were dispersed in 1.0 ml ethanol solution by sonication for 30 min to get a homogeneous ink. Then, certain amounts of the ink were loaded on a CFP (1 × 1 $cm^2$) and dried under infrared lamp. In Fig. 4, Pt or Ir metal loading was 0.1 $mg_{metal}$ $cm^{-2}$ to obtain appreciable catalytic activity. In the activity evaluation of M–$H_xWO_3$ systems, for parallel comparison, the loading amount of Ru/C, Pt/C and Pd/C comparison samples on CFP was 58 $μg_{Ru}$ $cm^{-2}$, 50 $μg_{Pt}$ $cm^{-2}$ and 65 $μg_{Pd}$ $cm^{-2}$, respectively. All potentials measured were calibrated vs RHE using the following equation: $E(RHE) = E(SCE) + 0.244 V + 0.0592 \times pH$. For each HER experiment, cathodic linear sweep voltammetry (LSV) was performed in high-purity $H_2$-saturated 1.0 M PBS (pH = 7.03 ± 0.08) or 0.5 M $H_2SO_4$ (pH = 0.35 ± 0.04) at a scan rate of 2 mV $s^{-1}$ (the pH value was determined by pH meter, see Supplementary Fig. 33). All the polarization curves were the steady ones after several scans with iR compensation. The iR compensation was adopted by on-the-fly correction with positive feedback mode, where the points were automatically corrected by instruments with built-in iR compensation. A factor of 95% was applied to measure the resistance value of electrolyte at the open circuit potential. Solution resistances in 1.0 M PBS were 0.4 ± 0.1, 0.6 ± 0.2, 0.8 ± 0.1, 1.1 ± 0.2, and 0.5 ± 0.2 ohm for commercial Pt/C, Ir/C, Ru/C, Pd/C and M–$H_xWO_3$ samples (M = Ir, Pt, Ru, Pd, Co, Ni), respectively. Solution resistances in 0.5 M $H_2SO_4$ were 0.3 ± 0.1, 0.5 ± 0.2, and 0.3 ± 0.1 ohm for commercial Pt/C, Ir/C and Ir–$H_xWO_3$ catalysts, respectively. The current density was calculated against the geometric area (1 $cm^2$) of the electrode to obtain the specific activity. Electrochemical impedance spectroscopy (EIS) measurements were performed from $10^{-2}$ to $10^5$ Hz with an amplitude of 5 mV at an overpotential of 30 mV in 1.0 M PBS solution. The stability measurements were performed by transient accelerating degradation technique (ADT) protocol and static chronopotentiometry test in 1.0 M PBS solution. ADT tests were conducted as follows. Typically, square-wave voltammetry consisting of 10,000 cycles were conducted between a higher potential of 0.15 $V_{RHE}$ and a lower potential of −0.35 $V_{RHE}$. Each cycle was maintained for 4 s. Chronopotentiometry was measured at 10 and 500 mA $cm^{-2}$ to represent typical static long-term stability tests.

## Measurement of pH on the catalyst surface

According to previous work[44], the pH values on the catalyst surfaces were measured by the rotating ring-disk electrode (RRDE) technique. The potential of Pt ring electrode (RE, 0.1866 $cm^{-2}$) is sensitive to pH and can be used to monitor the variations in the pH on the disk electrode (DE, 0.2475 $cm^{-2}$) surface. A three-electrode cell was constructed of the RRDE, graphite rod and a saturated calomel electrode (SCE) as working, counter, and reference electrodes, respectively. Then, the pH dependence of the open circuit potential ($E_{ocp}$) in $H_2$-saturated 1.0 M PBS solution was measured with Pt RE (Supplementary Fig. 5). The OCP of the Pt electrode would indicate the equilibrium potential of $2H^+ + 2e^- \rightarrow H_2$, which varies with pH according to the Nernst equation:

$$E(V \text{ vs. SHE}) = \frac{-2.303RT}{F}pH \tag{3}$$

The fugacity of $H_2$ is assumed to be equal to unity and $R$, $T$, and $F$ are the gas constant, the absolute temperature, and the Faraday constant.

For measuring the pH on the electrode surface, the investigated catalyst was loaded onto the disk electrode. The catalyst ink was prepared by ultrasonically dispersing catalyst powder (5 mg) in 5 wt% Nafion solution (20 μL) and ethanol (480 μL) mixed solution. 10 μL of catalyst ink (10.0 mg $mL^{-1}$) was then transferred onto the disk electrode (catalyst loading: 0.4 mg $cm^{-2}$). The pH measurements on the catalyst surfaces were conducted in 1.0 M PBS solution with the working electrode rotating at a speed of 1600 rpm. Constant potential method was performed on the disk electrode ($E = 0.1, 0, −0.1, −0.2, −0.3, −0.4, −0.5, −0.6,$ and $−0.7 V_{RHE}$ for 200 s) to obtain a steady-state

current response ($j$), and OCP was simultaneously measured on the Pt ring electrode. The pH value of the catalyst-loaded DE can be deducted from the pH value of the Pt RE by the following equation:

$$c_{\mathrm{rt,H^+}} - c_{\mathrm{rt,OH^-}} = N_{\mathrm{D}}(c_{\mathrm{d,H^+}} - c_{\mathrm{d,OH^-}}) + (1 - N_{\mathrm{D}})(c_{\infty,\mathrm{H^+}} - c_{\infty,\mathrm{OH^-}}) \quad (4)$$

where $c_{\mathrm{rt,H^+}}$ and $c_{\mathrm{d,H^+}}$ are the concentrations of $H^+$ on the RE and DE, respectively, $c_{\mathrm{rt,OH^-}}$ and $c_{\mathrm{d,OH^-}}$ are the concentrations of $OH^-$ on the RE and DE, respectively, $c_{\infty,\mathrm{H^+}}$ and $c_{\infty,\mathrm{OH^-}}$ are the concentrations of $H^+$ and $OH^-$ in the bulk electrolyte, respectively, and $N_{\mathrm{D}} = 0.37$ is the collection efficiency of the RE.

### Neutral water electrolysis device
For a neutral water electrolysis device system, the bifunctional Ir−$H_x$WO$_3$/CP catalysts ($2 \times 2$ cm$^2$) were both for the anodic OER and cathodic HER. As for benchmark commercial (−)Pt/C + Ir/C(+) partners, homogeneous slurries consisting of catalysts, Nafion solution (5.0 wt.%) and ethanol were air-sprayed onto the carbon fiber paper with an iridium black loading of 2.0 mg cm$^{-2}$ for the anode and 1.0 mg cm$^{-2}$ of Pt/C for the cathode. In all, 1.0 M PBS electrolyte was cycled both on the anodic and cathodic sides by a peristaltic pump, and the flow rate is 80 mL min$^{-1}$. Anion-exchange membrane (Fumasep FAA-3-PK-130) was used to isolate the cathode and anode. Polarization curves were collected from 1.0 to 3.5 V at room temperature under ambient pressure. The current density was calculated against the geometric area (4 cm$^2$) of the MEA to obtain the specific activity without iR compensation. The stability test was carried out by galvanostatic electrolysis at a constant current density of 150 mA cm$^{-2}$.

### Operando Raman spectroscopy measurement
Operando Raman spectra were recorded on a LabRAMHR Evolution with an Ar$^+$ laser of 514 nm excitation under controlled potentials by the electrochemical workstation. The electrolytic cell was homemade by Teflon with thin round quartz glass plate as cover to protect the objective. The Ir−$H_x$WO$_3$ and $H_x$WO$_3$ were directly used as working electrode. The Ag/AgCl electrode with inner reference electrolyte of 1.0 M KCl and a Pt wire serves as the reference electrode and the counter electrode, respectively. 1.0 M PBS was used as electrolyte and chronoamperometry measurements were conducted at the potential range from 0 to −0.30 V$_{\mathrm{RHE}}$ with the interval of 50 mV. The spectrum was obtained from at least 20 points to ensure accuracy of the information about the samples. The applied voltage–time of each point is 100 s, and the Raman test began when the time exceeded 60 s.

### Computational section
The Density Functional Theory (DFT) calculations were performed in the Vienna Ab initio simulation package (VASP) with the Perdew-Burke-Emzerhof (PBE). The projector augmented wave (PAW) functional was selected as the generalized gradient approximation (GGA) to describe the electron-ion interactions. The cut-off energy of 400 eV and Gaussian electron smearing method with $\sigma = 0.05$ eV were used. A vacuum of 15 Å was adopted along the $z$-axis. And ($5 \times 5 \times 1$) and ($2 \times 2 \times 1$) Monkhoest-Pack k-point mesh were used for all samples during electronic structure calculation and the structure optimization, respectively. During structure optimization, the geometry optimization was performed when the convergence criterion on forces became smaller than 0.02 eV Å$^{-1}$ and the energy difference was <10$^{-4}$ eV. To model the Ir−$H_x$WO$_3$ catalyst, the Ir$_{10}$ clusters were supported on $4 \times 4$ supercells of WO$_3$ (002) with surface O atoms saturated with H atoms. The atoms in the bottom two layers of WO$_3$ (002) were kept frozen while the remaining were allowed to relax during the slab calculations. Gibbs free energy of X species ($X = H_2O$ and H) adsorption is calculated by

$$\triangle G = \triangle E_{\mathrm{X/surf}} - \triangle E_{\mathrm{surf}} - \mathrm{u}E_{\mathrm{X}} + \triangle E_{\mathrm{ZPE}} - \mathrm{T}\triangle S_{\mathrm{H}} \quad (5)$$

where $E_{\mathrm{X/surf}}$ is the total energy of the surface with adsorbate, $E_{\mathrm{surf}}$ is the energy of the clean surface, $E_{\mathrm{X}}$ is the energy of adsorbate, $\Delta E_{\mathrm{ZPE}}$ represents the zero-point energy of the system, and $T\Delta S_{\mathrm{H}}$ is the contribution from entropy.

## Data availability
The authors declare that all data supporting the findings of this study can be found in the manuscript and Supplementary Information, or are available from the corresponding authors upon request.

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

## Acknowledgements

Financial support from the National Key R&D Program of China (2021YFB3801600), the "Pioneer" and "Leading Goose" R&D Program of Zhejiang (2023C01108, 2022C01218, and 2022C01151), and the National Natural Science Foundation of China (21872121 and 21908189) are greatly appreciated.

## Author contributions

X.Z. carried out most of the experimental work and wrote the manuscript. X.S. assisted the experimental work and discussed the results. X.Z., H.N., and R.Y. performed the DFT calculations. B.L., Q.L., S.M., and L.X. helped to polish the paper and discussed the results. Y.W. conceived the project and directed the overall work and manuscript writing. All the authors reviewed and contributed to this paper.

## Competing interests

The authors declare no competing interests.
