## [Peer Review File · Nature Communications]

REVIEWER COMMENTS

Reviewer #1 (Remarks to the Author):

Title: Tailing a local acid-like microenvironment for efficient neutral hydrogen evolution

In the manuscript by Xiaozhong Zheng et al., a novel Ir-HxWO₃ catalyst was synthesized and showed outstanding performance for HER in neutral media. By spontaneously injecting protons into WO₃, HxWO₃ was used as a proton sponge to create an acidic microenvironment around Ir sites.

Q.1 In order to take advantage of Hads' suitable adsorption energy on the cathode, the hydrogen evolution reaction (HER) takes place via a two-electron transfer process involving catalysts containing precious metals such as Pt and Pd. Could other precious metals have been used instead of iridium by the authors?

Q.2 With tungsten oxide as support, Chunlei Peng and colleagues (Appl. Catal. A, 623, 2021, 118293) have already reported improving HER performance by using Ir/W18O₄₉ nanowire catalysts. The authors should shed light on how this work differs from the reported work. Hence, in the present form I do not recommend the publication of this paper.

Minor:

Pg. No. 3, line 55, "as as a proton" replace with "as a proton"

Pg. No. 3, line 55, "a acid-like" instead of "an acid-like"

Pg. No. 22, line 426, "graphite rod" - should check font size

Reviewer #2 (Remarks to the Author):

In this work, the authors reported an interesting work of tailing the local catalytic microenvironment to improve the sluggish kinetics of neutral hydrogen evolution reaction (HER). Comprehensive experiments, ex/in-situ characterizations, and DFT calculations were employed to unveil the coherent synergism between Ir and spontaneously injected lattice-hydrogen species of HxWO₃. The Volmer process is drastically boosted at the Ir site to form Ir-H*, followed by thermodynamically favorable recombination of Ir-H* and neighboring revitalized WO-H* species to form H₂ molecular via interfacial Tafel step. The Ir-HxWO₃ shows exciting neutral HER activity with an ultralow overpotential of 20 mV at 10 mA cm⁻² and a low Tafel slope of 28 mV dec⁻¹, even comparable to those in an acidic environment. In summary, this work provides a novel and vital viewpoint to understand catalytic behavior in

electrochemical systems. The work is novel and important to the field. I recommend the acceptance of this work for publication in Nature Communications after the authors address the following revisions.

1. Whether the proposed concept or catalytic systems of this work can be transferred to other metal catalyst systems (such as Pt, Ru, etc.) and achieve similar effects?
2. Ir-based materials show high OER activity in acidic media. So, it would be very interesting if the author could test the OER activity of Ir-HxWO₃ in acidic solutions.
3. Generally, the Tafel slope for Pt/C in acidic media is about 30 mV dec⁻¹. In this work, the Tafel slope is ~15 mV dec⁻¹ in Figure 3b. Why? The authors should explain.
4. In figure 2d, the W 4f binding energies of the Ir-HxWO₃ shows a significant redshift compared with HxWO₃. It could be better if the author further provides the O 1s binding energies for both HxWO₃ and Ir-HxWO₃.
5. In Figure S6, it is not easy to identify the curves. It would be better if the authors use different colors.
6. The Ir-HxWO₃ shows outstanding HER activity in neutral conditions with an ultralow overpotential of 20 mV at 10 mA cm⁻², which shows good practical applications, while the stability test in Figure 3h is only nearly 50h. It would be better if the author can further extend the stability test time for such high-activity materials.
7. In the experimental section, the authors should state the details of accelerating the cyclic voltammetry cycling test to aid reproducibility.
8. Some important papers related to the local catalytic environment and/or HER are recommended to be cited: Nat Commun 13, 1189 (2022); Nat Commun 13, 5382 (2022); Nature Energy, 2023, doi.org/10.1038/s41560-023-01195-x; Nat. Mater. 2017,16, 57–69; Electrochemical Energy Reviews, 2022, 5(4), 1. <https://doi.org/10.1007/s41918-022-00136-8>; Nano-Micro Letters, 2022, 14, 120; Electrochem. Energy Rev. 2021, 4(3), 566-600; Small, 2022, 18, 2105803; SusMat, 2021, 1(4) : 460-481; etc.
9. There are a few mistakes in the manuscript, such as for Ir-HxWO₃ the title in table S2, the x and 3 should be subscript. The authors should check the typos in the manuscript.

Reviewer #3 (Remarks to the Author):

This paper introduced novel Ir nanoclusters embedded on locally hydrided WO₃ nanorods toward efficient neutral hydrogen evolution. Its superior HER activity and facile synthesis method are attractive for hydrogen production technology. However, material characterization and description of HER mechanism are not clear to explain superior HER activity. This reviewer cannot make a decision at this stage of process. Here are several specific comments, which should be addressed to improve the quality of this manuscript to meet the standard of the prestigious Nature Commun.

1. The author claimed that the local hydride WO₃ (HxWO₃) phase was created during the negative potential-cycling of WO₃ in PBS solution. Are there lattice contraction/extension after H insertion into WO₃?

2. About iridium:

- What does the meaning of "hexagonal Ir"? Metallic Ir has a cubic close packed crystal structure. The authors should improve the basic understanding of material chemistry.

- What is the loading amount of Ir in WO₃? The authors should evaluate the Ir loading with ICP method. The authors also provide Ir loading of commercialized Ir/C catalysts.

- Based on the TEM image in Fig 2b, the Ir loading seems not to be very low. Therefore, the authors must obtain the Ir crystal pattern from XRD analysis.

3. Typically Pt and Ru are well known as good HER catalysts. Why did the authors choose Ir for the neutral HER catalyst?

Reviewer #4 (Remarks to the Author):

This manuscript reports the application of Ir-HxWO₃ catalyst in the electrocatalytic hydrogen evolution reaction. The activity of the catalyst is high and a mechanism for HER was proposed, however, a similar catalyst and the idea of local acid-like microenvironment proposed to understand the HER have already been reported, therefore, the results described in this manuscript are not suitable for publication in Nature Communications. Additional comments are listed below for reference.

(1) The catalyst of Ir-HxWO₃ reported in this manuscript is similar to the previously reported noble metal-WO₃ HER catalyst, which reduces the novelty of this article (Nano Energy 71 (2020) 104653).

(2) The concept of creating a local acid-like microenvironment for HER has also been reported in a literature, which reduces the novelty of the article (NATURE COMMUNICATIONS | (2022) 13:2024; NATURE COMMUNICATIONS | (2019) 10:4876).

(3) The local acid-like microenvironment proposed by the authors lacks sufficient evidence. What is the pH of the local acid-like microenvironment? Is this local acid-like microenvironment stable during the HER processes?

(4) In the operando electrochemical Raman spectra, the WO-H in Ir-HxWO₃ gradually disappeared in the voltage range, and the explanation given by the authors is that the surface WO-H species were depleted

during the HER process. However, no obvious performance degradation was found in the long-term stability test, there exists a contradiction.

(5) In the XRD pattern of Ir-HxWO₃, Supplementary Figure 14a, why were the peaks of the carbon paper enhanced after the HER stability test?

(6) Li⁺ and SCN⁻ poisoning experiments for HxWO₃ and SCN⁻ poisoning experiments for Pt/C need to be carried to examine their HER mechanism proposed for HxWO₃.

Point-by-point response to the reviewers' comments

We sincerely thank the reviewer's for their careful review on our manuscript and their valuable comments and suggestions, which certainly help to improve our manuscript. We have revised and supplied a lot of experimental data and the corresponding explanations for improving our manuscript. All the changes have been marked in blue and highlighted in yellow in the revised manuscript and supplementary information. The point-by-point responses are presented below.

Comments by Reviewer's:

Reviewer #1 (Remarks to the Author): Title: Tailing a local acid-like microenvironment for efficient neutral hydrogen evolution. In the manuscript by Xiaozhong Zheng et al., a novel Ir-H_xWO₃ catalyst was synthesized and showed outstanding performance for HER in neutral media. By spontaneously injecting protons into WO₃, H_xWO₃ was used as a proton sponge to create an acidic microenvironment around Ir sites.

Response:

We express our sincere gratitude to the reviewer for your careful review on our manuscript and the constructive comments, which really help to improve the quality of our manuscript. Following the reviewer's suggestion. we have supplied a lot of experiment data to extent "proton sponges" effect of H_xWO₃ support to other catalytic systems (M-H_xWO₃, M = Pt, Ru, Pd, Ni, Co) for developing efficient neutral electrocatalysts, further proving the universality of the concept proposed by this paper. In addition, we have highlighted differences from other works in the revised manuscript to further demonstrate the quality and depth of this work. In general, we have made great changes to the manuscript according to the suggestions of reviewer, and we hope that our changes will satisfy the reviewer. Next, we will reply to the reviewer's comments one by one. All changes have been highlighted in the revised manuscript and supplementary information files as well.

Q.1 In order to take advantage of Hads' suitable adsorption energy on the cathode, the hydrogen evolution reaction (HER) takes place via a two-electron transfer process involving catalysts containing precious metals such as Pt and Pd. Could other precious

metals have been used instead of iridium by the authors?

Response:

We appreciate the reviewer’s meaningful suggestions. Inspired by the local acid-like microenvironment created by H_xWO_3 , it is expected to extend “proton sponge” effect of H_xWO_3 support to other $M-H_xWO_3$ systems ($M = Ru, Pt, Pd, Co, Ni$). As expected in Figure R2 and revised Supplementary Figure 28, compared with conventional M -Carbon catalysts, the significantly reduced overpotentials and accelerated reaction kinetics on $M-H_xWO_3$ systems ($M = Ru, Pt, Pd, Co, Ni$) confirm that the local acid-like microenvironment provided by H_xWO_3 fundamentally enhances the intrinsic HER activity of catalysts in thermodynamically unfavorable neutral media.

Figure R2 and revised Supplementary Figure 28. Neutral HER performance of $M-H_xWO_3$ and M -Carbon systems. The LSV curves, Tafel plots and corresponding activity comparison of M -

H_xWO_3 and M-Carbon systems (M = Ru, Pt, Pd, Co, Ni) in 1.0 M PBS solution. Ru: a-a2; Pt: b-b2; Pd: c-c2; Co: d-d2; Ni: e-e2.

Accordingly, the above highlighted text and corresponding figure are added to the **Main text** of the revised manuscript (in **Page 22, Line 428-434**) and supplementary information (in **Page S30**). The details for preparing M- H_xWO_3 materials are supplied in **Methods** section (in **Page 26, Line 506-514**) as below:

“Synthesis of M- H_xWO_3 (M = Pt, Ru, Pd, Ni and Co). M- H_xWO_3 (M = Ru, Pt and Pd) were synthesized by the same approach as the preparation of Ir- H_xWO_3 , except that metal precursor was changed to rutherfordium chloride, chloroplatinic acid or palladium chloride, respectively. Commercial 20 wt% Pt/C, 5 wt% Ru/C and 5 wt% Pd/C were used as comparison samples. M- H_xWO_3 (M = Ni and Co) were prepared as following steps. 400 μ L 20 g L⁻¹ nickel nitrate or cobalt nitrate was dropped onto 1×1 WO_3 /CFP, and then dried up. The resulting material was reduced in tube furnace at 500 °C for 3 h in H₂ atmosphere (flow rate: 50 sccm). The same approach was conducted on pure CFP to obtain Ni and Co comparison samples.”

Q.2 With tungsten oxide as support, Chunlei Peng and colleagues (*Appl. Catal. A*, 623, 2021, 118293) have already reported improving HER performance by using Ir/W₁₈O₄₉ nanowire catalysts. The authors should shed light on how this work differs from the reported work. Hence, in the present form I do not recommend the publication of this paper.

Response:

We sincerely thank the reviewer for the insightful comments. We have carefully read the paper (*Appl. Catal. A*, 623, 2021, 118293) provided by reviewer. Next, we will further emphasize the high quality and depth of this work by describing the differences between our work and the article on aspect of several perspectives, the special statements are as follows:

1) The determination of local acid-like microenvironment on H_xWO_3 support in neutral media.

We used rotating ring-disk electrode (RRDE) technique to quantitatively detect local pH on the H_xWO_3 cathode surfaces at different applied potentials in neutral 1.0 M PBS solution (bulk pH is 7.03), along with classical carbon support for comparison (Supplementary Fig. 4-6, details see Methods). On the basis of RRDE detecting method, we found that the pH of the H_xWO_3 surface varies from 6.27 to 3.53 as potential decreases from 0.1 to $-0.4 V_{RHE}$, which undoubtedly confirms that H_xWO_3 acts as proton sponge to form a local acid-like microenvironment on the electrode surface (Fig. 2a,b). In sharp contrast, the pH of carbon cathode surface is maintained at ~ 7 in range of $0.1 \sim -0.4 V_{RHE}$ (Fig. 2b). As the potential continued to shift negatively, the pH of H_xWO_3 and carbon support surfaces gradually increase and approach to 8.32 and 8.0 at $-0.7 V_{RHE}$, respectively, due to the consumption of the local hydrogen species. Astonishingly, when the bias ($-0.7 V_{RHE}$) is removed, the surface of H_xWO_3 and carbon cathodes turn back to steady acidic (pH = 3.73) and neutral (7.12) states, respectively (Fig. 2c).

Figure R3 and revised Fig. 2: Local pH measurements and catalytic behaviors of H_xWO_3 . a, Schematic diagram of local acid-like microenvironment generation on H_xWO_3 cathode. b, Measured pH values on H_xWO_3 and carbon cathode surfaces at different potentials, respectively. c, The pH on H_xWO_3 and carbon cathode surfaces before and after removing bias ($-0.7 V_{RHE}$). d, HER polarization curves of H_xWO_3 grown on CFP and (e) corresponding Tafel plots. f, Free energy barriers of H-H coupling and H-H transfer in H_xWO_3 .

Accordingly, the above highlighted text (in Page 6, Line 110-124) and

corresponding figure (in Page 8, Line 146-152) are added to the Main text of the revised manuscript.

2) Catalytic performances and stability.

As expected in Figure R4, although 10% Ir-W₁₈O₄₉ showed a competitive catalytic activity ($\eta_{10} = 83$ mV), it is still far behind the Ir-H_xWO₃ catalyst ($\eta_{10} = 20$ mV) in 1.0 M PBS solution. **This activity advantage is particularly evident under high current conditions.** Until now, developing high-current HER catalysts in neutral media remains a great challenge due to slow reaction kinetics and low ionic conductivity (*Adv. Energy Mater.*, 2023, 13, 2203164; *Adv. Mater.*, 2022, 34, 2200058; *ACS Sustainable Chem. Eng.*, 2021, 9, 11981; *ACS Catal.*, 2018, 8, 5200; *Energy Environ. Sci.*, 2020, 13, 3185). In Figure R4a and 4d, 10% Ir-W₁₈O₄₉ only provides a current density of nearly 30 mA/cm² at -0.2 V_{RHE}, while the current density of the Ir-H_xWO₃ catalyst is boosted by 11 times to achieve ~ 360 mA/cm² under the same potential conditions. Tafel slope plots further elucidate that Ir-H_xWO₃ (28 mV/dec) has faster kinetics compared with 10% Ir-W₁₈O₄₉ (66 mV/dec) in neutral-pH media (Figure R4b and 4e). With regards to catalytic stability (Figure R4c and 4f), amazingly, Ir-H_xWO₃ catalyst exhibits excellent operation stability at 10 and 500 mA cm⁻² over 100 h and 40 h, respectively, outperforming 10% Ir-W₁₈O₄₉, commercial Pt/C and most of the recently reported landmark catalysts (Supplementary Table 2). In summary, compared with 10% Ir-W₁₈O₄₉, the compelling activity and stability of Ir-H_xWO₃ catalyst make it more promising for applications in mild energy storage and conversion systems and direct seawater electrolysis for hydrogen production.

Figure R4. The activity and stability comparisons between 10% Ir- $W_{18}O_{49}$ and Ir- H_xWO_3 in 1.0 M PBS electrolyte. (a-c, *Appl. Catal. A*, 623, 2021, 118293) The LSV, Tafel slope and stability test of 10% Ir- $W_{18}O_{49}$ and associated catalysts; (d-f, this work) The LSV, Tafel slope and stability test of Ir- H_xWO_3 and associated catalysts.

3) Exploration of the catalytic mechanism of neutral HER.

Although much works have been done on tungsten oxide-supported nanometallic catalysts as excellent catalysts for energy conversion (*ACS Appl. Mater. Interfaces* 2017, 9, 31794; *ACS Appl. Mater. Interfaces* 2020, 12, 25991; *J. Mater. Chem. A*, 2019, 7, 6285; *Small*, 2021, 17, 2102159; *Appl. Catal. A*, 623, 2021, 118293; *Appl. Catal. B Environ.*, 2021, 296, 120359 and so on), the improved catalytic performances are simply attributed to the electronic regulation and synergistic catalytic effect. The exact catalytic mechanism is unclear, especially in challenging neutral media. There still lacks direct experimental evidence to prove the promoting effect of H_xWO_3 support in neutral HER. In this work, the solid experimental evidences that including *operando* Raman measurements, selective poisoning and kinetic isotope effect experiments confirm the coherent synergism between Ir and local Brønsted acid W-OH species of H_xWO_3 **for the first time**, that is, Volmer process (water dissociation step) is drastically boosted at Ir site to form Ir-H*, followed by spontaneous recombination of Ir-H* and neighboring revitalized WO-H* species to form H_2 molecular via interfacial Tafel step, as verified by theoretical calculations, thereby keeping the reaction at a high rate as observed in HER performance tests.

4) The extension of “proton sponges” effect of H_xWO_3 support.

In our work, the “proton sponge” effect of H_xWO_3 support can be universally transferred to other $M-H_xWO_3$ systems ($M = Ru, Pt, Pd, Co, Ni$). Compared with traditional M -Carbon systems, the significantly reduced overpotentials and accelerated reaction kinetics on $M-H_xWO_3$ systems ($M = Ru, Pt, Pd, Co, Ni$) confirm that the local acid-like microenvironment provided by H_xWO_3 fundamentally improve the intrinsic HER activity of catalysts in thermodynamically unfavorable neutral media.

5) Neutral water electrolysis device performance.

To highlight the practical significance of localized acidification environmental engineering for neutral water reduction, we further integrated bifunctional $Ir-H_xWO_3$ catalysts into a membrane electrode assembly (MEA) as cathode and anode materials and assembled an actual anion-exchange-membrane water electrolysis device (Figure R5 and Fig. 8a, details see Methods). The current density of the MEA composed of $Ir-H_xWO_3/CFP(\pm)$ is much higher than that of the MEA composed of benchmark commercial $(-)\text{Pt}/C + \text{Ir}/C(+)$ under the same cell voltage. At a current density of 10 mA cm^{-2} , the cell voltage is 1.78 V for $Ir-H_xWO_3/CFP(\pm)$ -based MEA system, which is much less than that of 2.05 V for benchmark commercial $(-)\text{Pt}/C + \text{Ir}/C(+)$ -based MEA setup (Fig. 8b). Significantly, the $Ir-H_xWO_3/CFP(\pm)$ MEA can be stably operated for at least 40 h at a larger current density of 150 mA cm^{-2} (Fig. 8c), demonstrating unprecedented application prospects.

Figure R5 and revised Fig. 8: Neutral water electrolysis device performance. **a**, Photographs and schematic illustration of membrane electrode assembly (MEA) electrochemical reactor, the geometric area of the electrode is 4 cm². **b**, Neutral water splitting performance of the commercial (-)Pt/C + Ir/C(+) and Ir-H_xWO₃/CFP(±) MEA setups at room temperature. **c**, Stability tests of the Ir-H_xWO₃/CFP(±) MEA.

Accordingly, the above highlighted text and corresponding figure are added to the **Main text** of the revised manuscript (in Page 23-24, Line 440-456).

Special changes in Introduction section are as follows:

In Page 3, Line 48:

“Therefore, selecting a suitable system to create a local acid-like environment through multiple physicochemical effects **to the maximum extent possible**, will provide an alternative way to promote the electrocatalytic performance and guide the higher efficiency electrocatalyst design in non-acidic electrolyte, especially in more challenging neutral media.”

In Page 3, Line 60-65:

“Up to now, **it still encounters many problems and challenges, for example, 1) the degree of local acidification of H_xWO₃ in neutral media has not been accurately quantified; 2) the synergistic catalysis between local acidic species and co-catalysts has not been fully understood; 3) the enhanced activity cannot be simply attributed to a**

single optimized catalytic site, and the origin of the activity deserves further investigation.”

Minor: Pg. No. 3, line 55, “as as a proton” replace with “as a proton” Pg. No. 3, line 55, “a acid-like” instead of “an acid-like” Pg. No. 22, line 426, “graphite rod”- should check font size.

Response:

We thank the reviewer for the useful suggestions. First of all, we are sorry for our carelessness. According to the kind reminders from reviewer, we have made corresponding changes in the revised manuscript, the details are as follows:

In Page 3, line 55:

“Protonated H_xWO_3 could act as a proton sponge and electron reservoir to create an acid-like microenvironment in the electric double layer, thereby further affecting the reaction barriers and pathway³⁴⁻³⁶.”

In Page 26, line 501:

“ WO_3 precursor, a graphite rod and a saturated calomel electrode (SCE) were used as the working electrode (WE), counter electrode (CE) and reference electrode (RE), respectively.”

Reviewer #2 (Remarks to the Author): In this work, the authors reported an interesting work of tailing the local catalytic microenvironment to improve the sluggish kinetics of neutral hydrogen evolution reaction (HER). Comprehensive experiments, ex/in-situ characterizations, and DFT calculations were employed to unveil the coherent synergism between Ir and spontaneously injected lattice-hydrogen species of H_xWO_3 . The Volmer process is drastically boosted at the Ir site to form Ir-H*, followed by thermodynamically favorable recombination of Ir-H* and neighboring revitalized WO_3-H^* species to form H_2 molecular via interfacial Tafel step. The Ir- H_xWO_3 shows exciting neutral HER activity with an ultralow overpotential of 20 mV at 10 mA cm^{-2} and a low Tafel slope of 28 mV dec^{-1} , even comparable to those in an acidic environment. In summary, this work provides a novel and vital viewpoint to understand catalytic behavior in electrochemical systems. The work is novel and important to the field. I recommend the acceptance of this work for publication in Nature Communications after the authors address the following revisions.

Response:

We appreciate the reviewer's positive feedbacks and evaluating our work as "novel and important to the field". We also express our sincere gratitude to the reviewer for all the constructive comments and suggestions, which really help to improve the quality of our manuscript. Following the reviewer's suggestions, some extended experiments were conducted to verify the universality of the proton sponge effect of H_xWO_3 support. In addition, we have cited some important literature that reviewer recommended and checked the manuscript carefully to avoid language errors. In generally, we hope that our changes will satisfy the reviewer. Next, we will reply to the reviewers' comments one by one. All changes have been highlighted in the revised manuscript and supplementary information files as well.

1. Whether the proposed concept or catalytic systems of this work can be transferred to other metal catalyst systems (such as Pt, Ru, etc.) and achieve similar effects?

Response:

We appreciate the reviewer's meaningful suggestions. Inspired by the local acid-

like microenvironment created by H_xWO_3 , it is expected to extend “proton sponge” effect of H_xWO_3 support to other $M-H_xWO_3$ systems ($M = Ru, Pt, Pd, Co, Ni$). As expected in Figure R2 and revised Supplementary Figure 28, compared with conventional M -Carbon catalysts, the significantly reduced overpotential and accelerated reaction kinetics on $M-H_xWO_3$ systems ($M = Ru, Pt, Pd, Co, Ni$) confirm that the local acid-like microenvironment provided by H_xWO_3 fundamentally improve the intrinsic HER activity of catalysts in thermodynamically unfavorable neutral media.

Figure R2 and revised Supplementary Figure 28. Neutral HER performance of $M-H_xWO_3$ and M -Carbon systems. The LSV curves, Tafel plots and corresponding activity comparison of $M-H_xWO_3$ and M -Carbon systems ($M = Ru, Pt, Pd, Co, Ni$) in 1.0 M PBS solution. Ru: a-a2; Pt: b-b2; Pd: c-c2; Co: d-d2; Ni: e-e2.

Accordingly, the above highlighted text and corresponding figure are added to the

Main text of the revised manuscript (in **Page 22, Line 428-434**) and supplementary information (in **Page S30**). The details for preparing M-H_xWO₃ materials are supplied in **Methods** section (in **Page 26, Line 506-514**) as below:

“Synthesis of M-H_xWO₃ (M = Pt, Ru, Pd, Ni and Co). M-H_xWO₃ (M = Ru, Pt and Pd) were synthesized by the same approach as the preparation of Ir-H_xWO₃, except that metal precursor was changed to rutherfordium chloride, chloroplatinic acid or palladium chloride, respectively. Commercial 20 wt% Pt/C, 5 wt% Ru/C and 5 wt% Pd/C were used as comparison samples. M-H_xWO₃ (M = Ni and Co) were prepared as following steps. 400 μL 20 g L⁻¹ nickel nitrate or cobalt nitrate was dropped onto 1×1 WO₃/CFP, and then dried up. The resulting material was reduced in tube furnace at 500 °C for 3 h in H₂ atmosphere (50 sccm). The same approach was conducted on pure CFP to obtain Ni and Co comparison samples.”

2. Ir-based materials show high OER activity in acidic media. So, it would be very interesting if the author could test the OER activity of Ir-H_xWO₃ in acidic solutions.

Response:

We appreciate the reviewer’s meaningful suggestions. According to the reviewer’s comments, we test the OER activity of Ir-H_xWO₃ in acidic and neutral solutions. As shown in the Figure R6, Ir-H_xWO₃ catalyst gives an extremely low OER overpotential of 247 and 327 mV at 10 mA cm⁻² and a Tafel slope of 87 and 70 mV dec⁻¹ in 0.5 M H₂SO₄ and 1 M PBS, respectively, both significantly lower than those of commercial Ir/C (acidic: 316 mV, 157 mV dec⁻¹; neutral: 497 mV, 247 mV dec⁻¹). Inspired by reviewer comments, Ir-H_xWO₃ is expected to act as a **“universally compatible”** electrocatalyst that simultaneously shows excellent HER and OER performances in neutral condition.

Figure R6. LSVs curve and Tafel slopes of of Ir-H_xWO₃ and commercial Ir/C catalysts in 0.5 M H₂SO₄ (a and b) and 1 M PBS solutions (c and d).

To highlight the practical significance of localized acidification environmental engineering for neutral water reduction, we further integrated bifunctional Ir-H_xWO₃ catalysts into a membrane electrode assembly (MEA) as cathode and anode materials and assembled an actual anion-exchange-membrane water electrolysis device (Figure R5 and Fig. 8a, details see Methods). The current density of the MEA composed of Ir-H_xWO₃/CFP(±) is much higher than that of the MEA composed of benchmark commercial (-)Pt/C + Ir/C(+) under the same cell voltage. At a current density of 10 mA cm⁻², the cell voltage is 1.78 V for Ir-H_xWO₃/CFP(±)-based MEA system, which is much less than that of 2.05 V for benchmark commercial (-)Pt/C + Ir/C(+)-based MEA setup (Fig. 8b). Significantly, the Ir-H_xWO₃/CFP(±) MEA can be stably operated for at least 40 h at a larger current density of 150 mA cm⁻² (Fig. 8c), demonstrating unprecedented application prospects.

Figure R5 and revised Fig. 8: Neutral water electrolysis device performance. **a**, Photographs and schematic illustration of membrane electrode assembly (MEA) electrochemical reactor, the geometric area of the electrode is 4 cm^2 . **b**, Neutral water splitting performance of the commercial $(-)\text{Pt}/\text{C} + \text{Ir}/\text{C}(+)$ and $\text{Ir}-\text{H}_x\text{WO}_3/\text{CFP}(\pm)$ MEA setups at room temperature. **c**, Stability tests of the $\text{Ir}-\text{H}_x\text{WO}_3/\text{CFP}(\pm)$ MEA.

Accordingly, the above highlighted text and corresponding figure are added to the **Main text** of the revised manuscript (in **Page 23-24, Line 440-456**). The details for neutral water electrolysis device was supplied in **Methods** section (in **Page 30-31, Line 604-616**) as below:.

Neutral water electrolysis device. For a neutral water electrolysis device system, the bifunctional $\text{Ir}-\text{H}_x\text{WO}_3/\text{CP}$ catalysts ($2 \times 2 \text{ cm}^2$) was both for the anodic OER and cathodic HER. As for benchmark commercial $(-)\text{Pt}/\text{C} + \text{Ir}/\text{C}(+)$ partners, homogeneous slurries consisting of catalysts, Nafion solution (5.0 wt.%) and ethanol were air-sprayed onto the carbon fiber paper with an iridium black loading of 2.0 mg cm^{-2} for the anode and 1.0 mg cm^{-2} of Pt/C for the cathode. In all, 1.0 M PBS electrolyte was cycled both on the anodic and cathodic sides by a peristaltic pump, and the flow rate is 80 mL min^{-1} . Anion-exchange-membrane (Fumasep FAA-3-PK-130) was used to isolate the cathode and anode. Polarization curves were collected from 1.0 to 3.5 V at room temperature under an ambient pressure. The current density was calculated against the geometric area (4 cm^2) of the MEA to obtain the specific activity without iR

compensation. The stability test was carried out by galvanostatic electrolysis at a constant current density of 150 mA cm⁻².”

3. Generally, the Tafel slope for Pt/C in acidic media is about 30 mV dec⁻¹. In this work, the Tafel slope is ~15 mV dec⁻¹ in Figure 3b. Why? The authors should explain.

Response:

We sincerely thank the reviewer for the insightful comments and we are pleased to clarify this issue. As we all known that, the Tafel equation is of fundamental importance in electrochemical kinetics, formulating a quantitative relation between the current and the applied electrochemical potential. Currently, Tafel plots are often extracted from potential polarization curves (such as linear sweep voltammetry and cyclic voltammetry) in the literature. However, the lack of a standard method for analyzing catalytic performance has prevented researchers from fairly comparing the catalytic properties of different nanomaterials for HER, even for state-the-of-art Pt/C catalytic system (*ACS Nano* 2018, 12, 9635-9638; *ACS Energy Lett.* 2021, 6, 1607-1611). Table R2 shows the comparison of the reported Tafel values of commercial Pt/C under acidic conditions. It can be seen that the Tafel values of commercial Pt/C range from 10 to 60 mV dec⁻¹ and are mostly below 30 mV dec⁻¹ (Figure R7-R10). These results reveal that the Tafel slope is strongly related to the scan rate, catalyst loading, substrate electrode, the selected fitting potential region and even manufacturer.

To address the reviewer’s concerns, we measured Tafel slopes of Pt/C on glassy carbon (GC) electrode again and verified the accuracy of our measurement. Firstly, to minimize the possibility of mass transport limitation associated with the reactant and product from the electrode, solution was stirred during the collection of Tafel data. Secondly, solution resistance was compensated (95%). Thirdly, a slow scan rate of 1 mV s⁻¹ was used to obtain the steady-state collection of current density at each applied potential. In addition, same loading amount of 0.1 mg_{Pt} cm⁻² was conducted to ensure a high activity. As shown in Figure R11, the Tafel slope of Pt/C is 13.2 mV dec⁻¹ in the range of 1 to 10 mA cm⁻², which is close to that of Pt/C on CFP. Combined with the above discussion, in this work, for the purpose of parallel performance comparison,

commercial 20 wt% Pt/C loading on carbon fiber paper was chosen as standard catalytic system.

Table R2. Comparison of acidic Tafel slopes of commercial Pt/C catalysts in recently reported literature.

Catalysts	Tafel (mV/dec)	Loading (mg _{Pt} /cm ²)	Electrolytes	Scan rate (mV/s)	References
Pt/C/CFP	15	0.1 on CFP	0.5 M H ₂ SO ₄	2	This work
Pt/C	22	0.051 on GC	0.5 M H ₂ SO ₄	5	Angew. Chem. Int. Ed. 2023, 62, e2023004.
Pt/C	12	0.025 on GC	0.5 M H ₂ SO ₄	10	Nat. Commun. 2022, 13, 5497.
Pt/C	12.4	0.0612 on GC	0.5 M H ₂ SO ₄	5	Angew. Chem. Int. Ed. 2021, 133, 12436-12442.
Pt/C	19.6	0.06 on GC	0.1 M HClO ₄	1	Nat Commun 2020, 11, 4246.
Pt/C	18.6	0.05 on GC	0.5 M H ₂ SO ₄	5	Nat. Commun. 2020, 11, 1029
Pt/C	23.6	-	0.5 M H ₂ SO ₄	10	Sci. China Mater. 2021, 64, 2467-2476.
Pt/C	27	0.057 on GC	0.5 M H ₂ SO ₄	5	Nat. nanotechnol. 2017, 12, 441.
Pt/C	29	0.08 on GC	0.5 M H ₂ SO ₄	1	Nat. Commun. , 2018, 9, 924.
Pt/C/CFP	34	0.0076 on CFP	0.5 M H ₂ SO ₄	50	Nat. Commun. , 2022, 13, 7225
Pt-C	47	0.0255 on GC	0.5 M H ₂ SO ₄	5	Adv. Mater. , 2021, 33, 2007894
Pt/C	24	0.0857 on GC	0.5 M H ₂ SO ₄	5	Adv. Mater. , 2021, 33, 2008599
Pt/C	26	0.0408 on GC	0.5 M H ₂ SO ₄	5	Angew. Chem. Int. Ed. 2019, 58, 16038-1604
Pt/C	25	0.2 on CP	0.5 M H ₂ SO ₄	1	Appl. Catal. B Environ. , 2022, 307,121199.
Pt/C	39	1.428 on GC	0.5 M H ₂ SO ₄	5	Small , 2023, 2301178.(10.1002/sml.202301178)
Pt/C	43	-	0.5 M H ₂ SO ₄	5	Adv. Energy Mater. 2023, 2300127.(10.1002/aenm.202300127)

Pt/C/CC	59	0.164 on CC	0.5 M H ₂ SO ₄	1	Nano Energy , 2017, 40, 27-33.
---------	----	----------------	---	---	--

Figure R7. (a) LSVs curve of of TiO₂, Pt/TiO₂, Pt/TiO₂-O_v and commercial Pt/C at 0.5 M H₂SO₄; (b) Tafel slope diagram of Pt/TiO₂, Pt/TiO₂-O_v and commercial Pt/C. (*Angew. Chem.Int. Ed.* 2023, 62, e2023004)

Figure R8. (a) LSV polarization curves with iR correction and (b) Tafel plots of Ru-1, Ru/RuS₂-2, Ru/RuS₂-4, RuS₂-8 and state-of-the-art Pt/C catalysts. (*Angew. Chem.*2021, 133, 12436-12442)

Figure R9. (a) HER polarization curves in 0.5 M H₂SO₄; (b) Comparison of the Tafel slope derived from polarization curves in acidic and alkaline electrolytes. (*Nat. Commun.* 2022, 13, 5497)

Figure R10. (a) LSV curves of the Pt₁/N-C, Pt/C and N-C framework; (b) Tafel plots for Pt₁/N-C and Pt/C electrocatalysts. (*Nat. Commun.* 2020, 11, 1029)

Figure R11. (a) LSV curve and (b) Tafel slope of the commercial Pt/C on GC (This work). All the measurements were operated in H₂-saturated 0.5 M H₂SO₄ with 95% iR -compensation. Catalyst loading: 0.1 mg_{Pt} cm⁻². scan rate: 1 mV s⁻¹.

4. In figure 2d, the W 4f binding energies of the Ir-H_xWO₃ shows a significant redshift compared with H_xWO₃. It could be better if the author further provides the O 1s binding energies for both H_xWO₃ and Ir-H_xWO₃.

Response:

Thank you for giving the constructive suggestions. According to the reviewer's comments. We supplied the high-resolution O 1s spectra of Ir-H_xWO₃ and H_xWO₃ samples. As shown in Figure R12 and revised Supplementary Figure 14, after loading Ir nanoparticles (NPs), the O 1s lineshape of Ir-H_xWO₃ catalyst is similar to that of H_xWO₃, indicating that the surfaces of H_xWO₃ and Ir-H_xWO₃ possess similar oxygen species. The split-peak fitting results reveals that the oxygen species on the H_xWO₃ and Ir-H_xWO₃ are lattice oxygen, surface hydroxyl and adsorbed water species, respectively. Notably, compared with H_xWO₃, the O 1s spectrum of Ir-H_xWO₃ catalysts moves to

lower binding energies, further illustrating the electron transfer process of Ir nanoparticles and H_xWO_3 support, which is consistent with the analysis of Ir 4f, W 4f and computational simulations (Fig. 3d-f).

Figure R12 and revised Supplementary Figure 14. Analysis of O 1s data. High-resolution O 1s spectra of Ir- H_xWO_3 and H_xWO_3 .

We have made corresponding changes, the details are as follows:

Main text. In Page 10, line 186-line 189:

“The O 1s split-peak fitting results (Supplementary Fig. 14) reveal that lattice oxygen, surface hydroxyl and adsorbed water species are detected on the surface of H_xWO_3 and Ir- H_xWO_3 samples. As expected, after Ir NPs loading, the O 1s spectrum of Ir- H_xWO_3 moves toward lower binding energies.”

5. In Figure S6, it is not easy to identify the curves. It would be better if the authors use different colors.

Response:

We thank the reviewer for the useful suggestions. Based on the suggestions of the reviewer, we replotted Supplementary Figure 9 with different colors to better present the experimental data.

Special changes are as follows:

In Page S11:

Figure R13 and revised Supplementary Figure 9. Activation process of Ir-H_xWO₃. HER polarization curves of the Ir-H_xWO₃ after different numbers of potential cycles were performed on it.

6. The Ir-H_xWO₃ shows outstanding HER activity in neutral conditions with an ultralow overpotential of 20 mV at 10 mA cm⁻², which shows good practical applications, while the stability test in Figure 3h is only nearly 50 h. It would be better if the author can further extend the stability test time for such high-activity materials.

Response:

Thank you for your constructive comments. According to your suggestion, we try to extend the stability time for Ir-H_xWO₃ catalyst at a constant current density of 10 mA/cm² and compare it with catalysts in the literature to highlight the stability of Ir-H_xWO₃ catalyst. As shown in the Figure R14 and revised Fig. 4h, increasing the constant current (@ 10 mA/cm²) electrolysis time to 100 hours, Ir-H_xWO₃ catalyst still represents a relatively stable horizontal line, and the overpotential of the catalyst before and after the test only increases by nearly 14 mV. Notably, the neutral HER catalytic stability is also superior to most of the recently reported landmark catalysts (Table R1 and revised Supplementary Table 2), especially under high current electrolysis. Ir-H_xWO₃ with high activity and high stability opens the door for the deployment of mild energy storage and conversion systems and direct seawater electrolysis for hydrogen production.

Figure R14 and revised Fig. 4h. Chronopotentiometry tests of Ir-H_xWO₃ (@10 and 500 mA/cm²) and commercial 20% Pt/C (@10 mA/cm²) in 1.0 M PBS.

Table R1 and Supplementary Table 2. Comparison of HER activities with various recently reported state-of-the-art catalysts in neutral electrolyte.

Catalysts	η_{10} /mV	Tafel /mV dec ⁻¹	Electrolyte	iR compensation	Stability	References
Ir-H_xWO₃	20	28	1 M PBS	95%	100 h @ 10 mA cm ⁻² 40 h @ 500 mA cm ⁻² 10,000 ADT cycles	This work
Ni_{0.1}Co_{0.9}P	125	109	1 M PBS	100%	20 h @ 30 mA cm ⁻²	Angew. Chem. Int. Ed. , 2018, 130, 15671
CrO_x/Ni-Cu	48	64	1 M PBS	100%	24 h @ -0.1 V _{RHE} ~ 33 mA cm ⁻²	Nat. Energy , 2019, 4, 107
N-Co₂P/CC	42	64	1 M PBS	100%	3000 CV cycles	ACS Catal. , 2019, 9, 3744
Pt_{SA}-NT-NF	24	30	1 M PBS	90%, R ~ 3 Ω	24 h @ 10 mA cm ⁻²	Angew. Chem. Int. Ed. , 2017, 56, 13694
Mn-Co-P/Ti	86	82	1 M PBS	100%	1000 CV cycles 10 h @ -0.096 V _{RHE} ~ 10 mA cm ⁻²	ACS Catal. , 2017, 7, 98
Ni(S_{0.5}Se_{0.5})₂	124	81	1 M PBS	-	2000 CV cycles 20 h @ -0.125 V _{RHE} ~ 10 mA cm ⁻²	J. Mater. Chem. A , 2019, 7, 16793
Ni_{0.89}Co_{0.11}Se₂	82	78	1 M PBS	-	40 h @ -0.2 V _{RHE} ~ 45 mA cm ⁻²	Adv. Mater. , 2017, 29, 1606521
MoP700	196	79	1 M PBS	iR -free	4000 CV cycles	ACS Catal. , 2019, 9, 8712

N-Ni	64	108	1 M PBS	100%	18 h @ 20 mA cm ⁻²	J. Am. Chem. Soc. , 2017, 139, 12283
Co-HNP/CC	85	38	1 M PBS	100%, $R \sim 1.7-2$ Ω cm ²	20 h @ 150 mA cm ⁻²	Angew. Chem., Int. Ed. , 2016, 55, 6725
CoP/Co-MOF	49	63	1 M PBS	100%	60,000 s @ 20 and 50 mA cm ⁻²	Angew. Chem., Int. Ed. , 2019, 58, 4679
Karst NF	110	99	-	90%	10 h @ -0.21 V _{RHE} ~ 10 mA cm ⁻²	Energy Environ. Sci. , 2020, 13, 174
Ni _{0.33} Co _{0.67} S ₂	72	68	1 M PBS	100%	20 h @ -0.07 V _{RHE} ~ 10 mA cm ⁻²	Adv. Energy Mater. , 2015, 5, 1402031
Cu _{0.08} Co _{0.92} P	81	83.5	0.5 M KHCO ₃	100%	3000 CV cycles 20 h @ 10 mA cm ⁻²	Appl. Catal., B. 2020, 265, 118555
Pt/np-Co _{0.85} Se	55	35	1 M PBS	100%	40 h @ -0.05 V _{RHE} ~ 10 mA cm ⁻²	Nat. Commun. 2019, 10, 1743

Special changes are as follows:

1) We have updated Fig. 4h (In Page 15-16) and Supplementary Table 2 (In Page S32-S33) in the revised manuscript and supplementary information.

2) Main text. In Page 14, line 280-287:

“The long-term durability testing on the Ir-H_xWO₃ catalyst by a static chronopotentiometry test (at 10 mA cm⁻² in Fig. 4h) even represents a relatively stable horizontal line with an overpotential increase of only 14 mV over 100 operating hours, while Pt/C exhibits a clipping activity decay within a few hour, demonstrating the outstanding activity retention of Ir-H_xWO₃. However, it is still a challenge to develop efficient catalysts working at large current density for neutral water splitting. Amazingly, Ir-H_xWO₃ catalyst maintains operation stability at a large current density of 500 mA cm⁻² over 40 h, outperforming most of the recently reported landmark catalysts (Supplementary Table 2).”

7. In the experimental section, the authors should state the details of accelerating cyclic

voltammetry cycling test to aid reproducibility.

Response:

We are sorry for our carelessness. According to the reviewer's suggestion, some important details on accelerating degradation technique was provided in experimental section.

Special changes are as follows:

Methods section. In Page 29, line 569-line 574:

“The stability measurements were performed by transient accelerating degradation technique (ADT) protocol and static chronopotentiometry test in 1.0 M PBS solution. ADT tests were conducted as follows. Typically, square-wave voltammetry consisted of 10,000 cycles were conducted between a higher potential of 0.15 V_{RHE} and an lower potential of -0.35 V_{RHE}. Each cycle was maintained for 4 s. Chronopotentiometry was measured at 10 and 500 mA cm⁻² to represent typical static long-term stability tests.”

8. Some important papers related to the local catalytic environment and/or HER are recommended to be cited: Nat Commun 13, 1189 (2022); Nat Commun 13, 5382 (2022); Nature Energy, 2023, doi.org/10.1038/s41560-023-01195-x; Nat. Mater. 2017,16, 57–69; Electrochemical Energy Reviews, 2022, 5(4), 1. <https://doi.org/10.1007/s41918-022-00136-8>; Nano-Micro Letters, 2022, 14, 120; Electrochem. Energy Rev. 2021, 4(3), 566-600; Small, 2022, 18, 2105803; SusMat, 2021, 1(4): 460-481; etc.

Response:

Thank you for this kind suggestion. We have read these papers carefully, and cited some important papers as Ref [4-7, 13, 19, 20] in the revised manuscript. The specific changes are as follows.

The added References. In Page 33, line 659-line 669:

4. Wang, P. et al. Interface engineering of Ni_xS_y@ MnO_xH_y nanorods to efficiently enhance overall-water-splitting activity and stability. *Nano-Micro Lett.* **14**, 120 (2022).
5. Xu, X., Sun, H., Jiang, S. P. & Shao, Z., Modulating metal-organic frameworks for catalyzing acidic oxygen evolution for proton exchange membrane water electrolysis. *SusMat* **1**, 460-481 (2021).

6. Ali, A., Long, F. & Shen, P. K., Innovative Strategies for Overall Water Splitting Using Nanostructured Transition Metal Electrocatalysts. *Electrochem. Energy Rev.* **5**, 1 (2022).

7. Wang, P. et al. MnO_x-Decorated Nickel-Iron Phosphides Nanosheets: Interface Modifications for Robust Overall Water Splitting at Ultra-High Current Densities. *Small* **18**, e2105803 (2022).

In Page 34, line 682-line 683:

13. Dai, J. et al. Hydrogen spillover in complex oxide multifunctional sites improves acidic hydrogen evolution electrocatalysis. *Nat. Commun.* **13**, 1189 (2022).

In Page 34-35, line 696-line 699:

19. Guo, J. et al. Direct seawater electrolysis by adjusting the local reaction environment of a catalyst. *Nat. Energy* **8**, 264-272 (2023).

20. Stamenkovic, V., Strmcnik, D., Lopes, P. & Markovic, N. M. et al. Energy and fuels from electrochemical interfaces. *Nat. Mater.* **16**, 57-69 (2017).

9. There are a few mistakes in the manuscript, such as for Ir-H_xWO₃ the title in table S2, the x and 3 should be subscript. The authors should check the typos in the manuscript.

Response:

We thank the reviewer for the useful suggestions. First of all, we are sorry for our carelessness. According to the kind reminders from reviewer, we have made corresponding changes in the revised supplementary information. Meanwhile, we also check the revised manuscript thoroughly to avoid the aforementioned problems.

In Page S31:

Supplementary Table 1. Noble metal content of M-H_xWO₃ (M = Ir, Ru, Pt, Pd), Ir/C, Pt/C, Ru/C and Pd/C determined by inductively coupled plasma optical emission spectrometry (ICP-OES).

Catalysts	Initial	After long-term test
Ir-H _x WO ₃ /CFP	^a 2.8%	^a 2.6%

	^b 47 $\mu\text{g cm}^{-2}$	^b 40 $\mu\text{g cm}^{-2}$
Ru-H _x WO ₃ /CFP	58 $\mu\text{g cm}^{-2}$	/
Pt-H _x WO ₃ /CFP	50 $\mu\text{g cm}^{-2}$	/
Pd-H _x WO ₃ /CFP	65 $\mu\text{g cm}^{-2}$	/
Commercial 10 wt% Ir/C	9.8 %	/
Commercial 20 wt% Pt/C	19.6 %	/
Commercial 5 wt% Ru/C	4.9 %	/
Commercial 5 wt% Pd/C	4.9 %	/

^a Ir loading normalized to H_xWO₃ support, the calculation formula is as follows: Ir wt% =

$$\frac{m(\text{Ir})}{m(\text{Ir-H}_x\text{WO}_3/\text{CFP})-m(\text{CFP})} * 100 \%$$

^b Ir loading normalized to geometric area of CFP support, the calculation formula is as follows: Ir

$$\text{wt}\% = \frac{m(\text{Ir})}{A(\text{CFP})}$$

Reviewer #3 (Remarks to the Author): This paper introduced novel Ir nanoclusters embedded on locally hydrided WO_3 nanorods toward efficient neutral hydrogen evolution. Its superior HER activity and facile synthesis method are attractive for hydrogen production technology. However, material characterization and description of HER mechanism are not clear to explain superior HER activity. This reviewer cannot make a decision at this stage of process. Here are several specific comments, which should be addressed to improve the quality of this manuscript to meet the standard of the prestigious Nature Commun.

Response:

We express our sincere gratitude to the reviewer for all the constructive comments and suggestions, which really helped to improve the quality of our manuscript. Following the reviewer's suggestions, we determine the Ir loading amount of Ir- H_xWO_3 and commercial Ir/C catalysts, and have updated the descriptions about Ir crystal structure. In addition, we experimentally and theoretically elucidate why Ir was chosen as the active component in this work. In generally, we hope that our changes will satisfy the reviewer. Next, we will reply to the reviewers' comments one by one. All changes have been highlighted in the revised manuscript and supplementary information files as well.

1. The author claimed that the local hydride WO_3 (H_xWO_3) phase was created during the negative potential-cycling of WO_3 in PBS solution. Are there lattice contraction/extension after H insertion into WO_3 ?

Response:

We thank the reviewer for the useful suggestions. As the reviewer mentioned that, the injection of foreign hydrogen atoms inevitably affects the lattice structure of the WO_3 host. The characteristic diffraction peak of WO_3 in H_xWO_3 and Ir- H_xWO_3 catalysts are shifted toward low-angle direction, as can be seen in a small range of XRD pattern (Figure R15 and revised Supplementary Figure 10), probably ascribing to oxygen defect-induced lattice extension (*Adv. Mater.* 2019, 31, 1903738; *Ceram. Int.*

2021, 47, 5091).

Figure R15 and Supplementary Figure 10. Analysis of XRD data. The XRD patterns of WO₃, H_xWO₃ and Ir-H_xWO₃ catalysts in the small range.

Accordingly, we added Figure R15 as Supplementary Figure 10 in the revised supplementary information (in **Page S12**). we have changed the statements in the revised manuscript, the specific as follows:

Main text. In Page 9, line 164-line 167:

“Notably, the characteristic diffraction peak of WO₃ in H_xWO₃ and Ir-H_xWO₃ catalysts are shifted toward low-angle direction (Supplementary Fig. 10), probably ascribing to oxygen defect-induced lattice extension⁴⁷”

The added References. In Page 38, line 763-line 764:

47. Ning, S. et al. Anomalous defect dependence of thermal conductivity in epitaxial WO₃ thin films. *Adv. Mater.* **31**, 903738 (2019).

2. About iridium: - What does the meaning of “hexagonal Ir”? Metallic Ir has a cubic close packed crystal structure. The authors should improve the basic understanding of material chemistry. - What is the loading amount of Ir in WO₃? The authors should evaluate the Ir loading with ICP method. The authors also provide Ir loading of commercialized Ir/C catalysts. - Based on the TEM image in Fig 2b, the Ir loading seems not to be very low. Therefore, the authors must obtain the Ir crystal pattern from XRD analysis.

Response:

(1) We thank the reviewer for raising the professional question. First of all, we apologize for confusing reviewers with our incorrect descriptions for Ir crystal structure. In fact, the reviewer is correct. As shown in the Figure R16 and revised Fig. 2c, the fast Fourier transform (FFT) pattern of Ir nanoparticle exhibits (111) and (200) facets, suggesting face-centered cubic crystal structure, which also agree with the results of the literature (*Nat. Commun.*, 2021, 12, 4271; *Adv. Funct. Mater.*, 2022, 32, 2113191; *Adv. Mater.*, 2018, 30, 1805606; *Appl. Catal., B*, 2019, 258, 117965) and crystal library data (<https://materialsproject.org/>). We accordingly modified Fig. 3c and the statements of the article, the specific as follows:

Main text. In Page 9, line 175-line 178:

“The Ir NPs exhibit clear lattice fringes with interplanar distances of 1.97 and 2.28 Å, corresponding to the [200] and [111] crystal planes of the the [011] face-centered cubic Ir, respectively, further evidenced by the fast Fourier transform (FFT) pattern (Fig. 3c).”

Figure R16 and revised Fig. 3c. The typical HRTEM images of Ir-H_xWO₃, with the inset is the fast Fourier transform image of the corresponding Ir NP.

Main text. In Page 12:

Fig. 3: Structural characteristics of catalysts. **a**, XRD patterns of as-obtained WO_3 , H_xWO_3 and $\text{Ir-H}_x\text{WO}_3$. **b**, Typical TEM image of $\text{Ir-H}_x\text{WO}_3$, the inset in **(b)** shows the Ir NPs size distribution pattern. **c**, HAADF-STEM images of $\text{Ir-H}_x\text{WO}_3$, with the inset in **(c)** giving the fast Fourier transform of the corresponding Ir NP. **d**, High-resolution W 4f spectra of $\text{Ir-H}_x\text{WO}_3$ and H_xWO_3 . **e**, High-resolution Ir 4f spectra of $\text{Ir-H}_x\text{WO}_3$ and commercial Ir/C. **f**, The side views of charge density difference in the interface of Ir_{10} clusters supported on H_xWO_3 . The cyan region reflects an electron-deficient state while the yellow region reflects an electron-rich area, with an isovalue of $0.004 \text{ e}^-/\text{\AA}^3$. **g**, Representative TPD-MS thermal desorption profiles for $\text{Ir-H}_x\text{WO}_3$ and H_xWO_3 and the main desorbed specie detected is H_2 with $m/e = 2$. **h**, Electron localization function evaluations of $\text{Ir}_{10}\text{-H}_x\text{WO}_3$. **i**, The pDOS curves of 1s orbitals of different hydrogen atoms in $\text{Ir}_{10}\text{-H}_x\text{WO}_3$ and pure H_xWO_3 .

(2) According to the review's suggestions, we determined the loading amount of Ir on H_xWO_3 support with inductively coupled plasma-optical emission spectrometry (ICP-OES). The specific results are shown in the Table R3 and revised Supplementary Table 1.

Table R3 and revised Supplementary Table 1. Noble metal content of $\text{M-H}_x\text{WO}_3$ ($\text{M} = \text{Ir, Ru, Pt, Pd}$), Ir/C, Pt/C, Ru/C and Pd/C determined by inductively coupled plasma optical emission spectrometry (ICP-OES).

Catalysts	Initial	After long-term test
-----------	---------	----------------------

	^a 2.8%	^a 2.6%
Ir-H _x WO ₃ /CFP	^b 47 μg cm ⁻²	^b 40 μg cm ⁻²
Ru-H _x WO ₃ /CFP	58 μg cm ⁻²	/
Pt-H _x WO ₃ /CFP	50 μg cm ⁻²	/
Pd-H _x WO ₃ /CFP	65 μg cm ⁻²	/
Commercial 10 wt% Ir/C	9.8 %	/
Commercial 20 wt% Pt/C	19.6 %	/
Commercial 5 wt% Ru/C	4.9 %	/
Commercial 5 wt% Pd/C	4.9 %	/

^a Ir loading normalized to H_xWO₃ support, the calculation formula is as follows: Ir wt% =

$$\frac{m(\text{Ir})}{m(\text{Ir-H}_x\text{WO}_3/\text{CFP})-m(\text{CFP})} * 100 \%$$

^b Ir loading normalized to geometric area of CFP support, the calculation formula is as follows: Ir

$$\text{wt}\% = \frac{m(\text{Ir})}{A(\text{CFP})}$$

Accordingly, we have changed the statements in the revised manuscript, specific as follows:

Main text. In Page 8, line 159-line 161:

“Ir content in Ir-H_xWO₃ hybrid electrocatalyst is 2.8 wt% (or 47 μg cm⁻² when normalized to geometric area of electrode) determined by inductively coupled plasma-optical emission spectrometry (ICP-OES, Supplementary Table 1).”

(3) We provide additional larger-scale TEM images to reflect the distribution of Ir NPs in H_xWO₃ nanorods support. As displayed in the typical TEM images of Ir-H_xWO₃ at different locations (Figure R17), Ir NPs are sparsely loaded on H_xWO₃. Based on that, the XRD signals of Ir NPs are not detected, possibly due to their small sizes (~ 1.7 nm) and low content (2.8 wt%), which is consistent with noble-metal-supported

catalysts in the reported literature (*ACS Catal.*, 2017, 7, 7131; *Energy Environ. Sci.*, 2018, 11, 800; *Angew. Chem. Int. Ed.*, 2023, 62, e2023022). In order to further eliminate the doubts of reviewer, different Ir loading amount (3, 5, 10 and 20 wt%) on H_xWO_3 support catalysts were prepared to obtain their XRD patterns. As we can see in the Figure R18, the characteristic diffraction peak of metal Ir (PDF#02-1155) did not appear until the Ir loading reaches 20%.

Figure R17. The typical TEM images of Ir- H_xWO_3 at different locations.

Figure R18. The XRD patterns of a series Ir- H_xWO_3 catalysts with different Ir loading.

3. Typically Pt and Ru are well known as good HER catalysts. Why did the authors choose Ir for the neutral HER catalyst?

Response:

We sincerely thank the reviewer for the insightful comments and we are pleased to clarify this issue. Then, the detailed interpretations are presented from both the experimental and theoretical aspects as listed below:

1) From theoretical aspect:

Iridium (Ir), with its weak hydrogen binding energy, was first selected for the chemisorption of hydrogen. In a hydrogen adsorption/desorption test, Ir with a specific (111) facet is thermally stable and exhibits relatively balanced hydrogen adsorption/desorption capacity, comparable to Pt(100) (*J. Chem. Phys.* **1987**, 87, 3104). For HER, a classical volcano-shaped correlation by J. K. Nørskov (*J. Electrochem. Soc.*, 2005, 152, J23) is found from both experimental results as well as computational approaches with metals that adsorb hydrogen neither strongly nor weakly (the Pt group metals) occupying the apex of the volcano curve (Figure R19). While the metals that adsorb hydrogen strongly (Ru and 3d elements) are positioned on the descending part of the volcano plot (*Angew. Chem. Int. Ed.* 2012,124, 12663). Therefore, its position near the top of the HER volcano plot ensures that Ir-based HER electrocatalysts receive considerable attention as a viable alternative to Pt (*Nat. Commun.*, 2019, 10, 4060; *Nat. Commun.*, 2020, 11, 4246; *Adv. Energy Mater.*, 2018, 8, 1801698). However, under neutral condition, the adsorption/activation of interfacial water molecular is also a crucial step in HER process (*Adv. Energy Mater.* 2023, 13, 2203164; *Energy Environ. Sci.*, 2020, 13, 3185). Compared with Pt, higher oxygenophilic properties of Ir and Ru confer a lower water dissociation energy barrier, thus providing hydrogen species for subsequent H₂ generation. Strmcnik, D. et al. found that the alkaline HER/HOR activities for Ir(111), polycrystalline Ir and a Pt-Ru alloy were much higher than those for Pt(111), Au(111) and Ru(0001) owing to different OH adsorption capacity (*Nat. Chem.* 2013, 5, 300). In summary, from a theoretical point of view, **the moderate hydrogen adsorption strength and strong oxygenophilic feature of Ir** make it a promising candidate for neutral HER catalysts and can be adopted as an ideal model to study the chemisorption of hydrogen during the HER process regardless of its high price.

Figure R19. The typical Volcano diagram of exchange current densities as a function of ΔG_{H^*} proposed by J. K. Nørskov (*J. Electrochem. Soc.*, 2005, 152, J23).

2) From experimental aspect:

Furthermore, we synthesized Pt- H_xWO_3 and Ru- H_xWO_3 catalysts and compared them with Ir- H_xWO_3 , respectively. As shown in the Figure R20, Ir- H_xWO_3 catalyst exhibits lowest overpotentials at different current densities and significantly expedited reaction kinetics. More importantly, in addition to catalytic activity, the **superior stability** (100 h@10 mA cm⁻² with $\Delta\eta = 14$ mV) makes Ir- H_xWO_3 catalysts more promising for industrial applications compared with Pt- H_xWO_3 (40 h@10 mA cm⁻² with $\Delta\eta = 132$ mV) and Ru- H_xWO_3 (48 h@10 mA cm⁻² with $\Delta\eta = 76$ mV). Additionally, Ir- H_xWO_3 catalyst gives an extremely low OER overpotential of 327 mV at 10 mA cm⁻² (Figure R21) and a Tafel slope of 70 mV dec⁻¹ in and 1 M PBS, both significantly lower than those of commercial Ir/C (497 mV, 247 mV dec⁻¹). Therefore, Ir- H_xWO_3 is expected to act as a “**universally compatible**” electrocatalyst that simultaneously shows excellent HER and OER performances in neutral condition.

Figure R20. (a) The LSV plots of Ir-H_xWO₃, Ru-H_xWO₃ and Pt-H_xWO₃ catalysts in 1.0 M PBS electrolyte; (b) Activity comparisons of catalysts; (c) Tafel plots of catalysts; (d) The stability test at 10 mA cm⁻².

Figure R21. The LSV and Tafel plots of Ir-H_xWO₃ and commercial Ir/C catalysts in 1.0 M PBS electrolyte.

To highlight the practical significance of localized acidification environmental engineering for neutral water reduction, we further integrated bifunctional Ir-H_xWO₃ catalysts into a membrane electrode assembly (MEA) as cathode and anode materials and assembled an actual anion-exchange-membrane water electrolysis device (Figure R5 and Fig. 8a, details see Methods). The current density of the MEA composed of Ir-H_xWO₃/CFP(±) is much higher than that of the MEA composed of benchmark commercial (-)Pt/C + Ir/C(+) under the same cell voltage. At a current density of 10 mA cm⁻², the cell voltage is 1.78 V for Ir-H_xWO₃/CFP(±)-based MEA system, which

is much less than that of 2.05 V for benchmark commercial (-)Pt/C + Ir/C(+)-based MEA setup (Fig. 8b). Significantly, the Ir-H_xWO₃/CFP(±) MEA can be stably operated for at least 40 h at a larger current density of 150 mA cm⁻² (Fig. 8c), demonstrating unprecedented application prospects.

Figure R5 and revised Fig. 8: Neutral water electrolysis device performance. a, Photographs and schematic illustration of membrane electrode assembly (MEA) electrochemical reactor, the geometric area of the electrode is 4 cm². b, Neutral water splitting performance of the commercial (-)Pt/C + Ir/C(+) and Ir-H_xWO₃/CFP(±) MEA setups at room temperature. c, Stability tests of the Ir-H_xWO₃/CFP(±) MEA.

Accordingly, the above highlighted text and corresponding figure are added to the Main text of the revised manuscript (in Page 23-24, Line 440-456). The details for neutral water electrolysis device are supplied in Methods section (in Page 30-31, Line 604-616) as below:.

Neutral water electrolysis device. For a neutral water electrolysis device system, the bifunctional Ir-H_xWO₃/CP catalysts (2×2 cm²) was both for the anodic OER and cathodic HER. As for benchmark commercial (-)Pt/C + Ir/C(+) partners, homogeneous slurries consisting of catalysts, Nafion solution (5.0 wt.%) and ethanol were air-sprayed onto the carbon fiber paper with an iridium black loading of 2.0 mg cm⁻² for the anode and 1.0 mg cm⁻² of Pt/C for the cathode. In all, 1.0 M PBS electrolyte was cycled both on the anodic and cathodic sides by a peristaltic pump, and the flow rate is 80 mL min⁻¹.

¹. Anion-exchange-membrane (Fumasep FAA-3-PK-130) was used to to isolate the cathode and anode. Polarization curves were collected from 1.0 to 3.5 V at room temperature under an ambient pressure. The current density was calculated against the geometric area (4 cm²) of the MEA to obtain the specific activity without *iR* compensation. The stability test was carried out by galvanostatic electrolysis at a constant current density of 150 mA cm⁻².”

Reviewer #4 (Remarks to the Author): This manuscript reports the application of Ir- H_xWO_3 catalyst in the electrocatalytic hydrogen evolution reaction. The activity of the catalyst is high and a mechanism for HER was proposed, however, a similar catalyst and the idea of local acid-like microenvironment proposed to understand the HER have already been reported, therefore, the results described in this manuscript are not suitable for publication in Nature Communications. Additional comments are listed below for reference.

We express our sincere gratitude to the reviewer for your careful review on our manuscript and the constructive comments, which really helped to improve the quality of our manuscript. Following the reviewer's suggestions, we use rotating ring-disk electrode (RRDE) technique to quantitatively detect local pH on the H_xWO_3 cathode surfaces at different applied potentials in neutral 1.0 M PBS solution (bulk pH is 7.03), along with classical carbon support for comparison. In addition, we have highlighted differences from other works that the reviewer provided to further demonstrate the novelty and depth of this work. In general, we have made great changes to the manuscript according to the suggestions of reviewer, and we hope that our changes will satisfy the reviewer. Next, we will reply to the reviewers' comments one by one. All changes have been highlighted in the revised manuscript and supplementary information files as well.

(1) The catalyst of Ir- H_xWO_3 reported in this manuscript is similar to the previously reported noble metal- WO_3 HER catalyst, which reduces the novelty of this article (Nano Energy 71 (2020) 104653). The concept of creating a local acid-like microenvironment for HER has also been reported in a literature, which reduces the novelty of the article (NATURE COMMUNICATIONS | (2022) 13:2024; NATURE COMMUNICATIONS | (2019) 10:4876).

Response:

We sincerely thank the reviewers for their comments on this work and we are happy to reemphasize the high quality and depth of this work by describing the differences between our work and previous works provided by reviewer on aspect of

several perspectives, the specific statements are as follows:

(1) Nano Energy 71 (2020) 104653.

1.1 Differences in electrolyte systems.

Wang et al. controllably synthesized a series of tungsten oxides loaded platinum catalysts (Pt-WO₃) with excellent HER activity closing to that of commercial Pt/C **in acidic media** (0.5 M H₂SO₄ solution). It is well accepted that the presence of a large number of H₃O⁺ species in the acidic solution allows the Volmer step to proceed smoothly on most of the catalyst surfaces (M-H_{ads}), followed by hydrogen generation *via* Heyrovsky or Tafel step. The metallic platinum (Pt) is still considered as “the Holy Grail” of HER electrocatalyst in acidic media with a nearly-zero onset overpotential and fast kinetics owing to its favorable hydrogen binding energy. Excellent catalytic activity is usually obtained with Pt-based catalysts under acidic media (*Angew. Chem. Int. Ed.*, 2023, 135, e202300; *Appl. Catal., B*, 2022, 314, 121503; *J. Electroanal. Chem.*, 2021, 896, 115076). Therefore, high intrinsic activity of Pt makes the WO₃ support less prominent. In principle, unlike the single-step reduction (Volmer reaction) of H₃O⁺ ions or H₂O molecules taking place in strong acidic or alkaline electrolytes, respectively, HER in neutral/near-neutral electrolytes has two overwhelming features, including the reactant (H₃O⁺ ions and H₂O molecules) switching and the diffusion-controlled kinetics, making the development of efficient neutral HER catalysts sharply challenging. In our work, Ir-H_xWO₃ was synthesized and functioned as **efficient neutral HER catalyst**, demonstrating acid-like activity with ultralow overpotential of 20 mV at 10 mA cm⁻² and low Tafel slope of 28 mV dec⁻¹ in 1 M PBS solution, which is even comparable to those in acidic environment. Owing to the local acid microenvironment induced by H_xWO₃ support in reaction condition, the Ir-H_xWO₃ catalyst breaks the traditional pH-dependent kinetics limitations compared with conventional Ir/C and Pt/C systems. Therefore, our work is not only significant for adding new knowledge to the basic hydrogen evolution catalysis, but also essential for the design and development of neutral HER electrocatalysts for mild energy conversion systems.

1.2 Differences in catalytic mechanisms.

Wang et al. claimed that metallic **Pt⁰ clusters were identified as the real catalytic active sites**. the electrochemically driven from WO_3 to H_xWO_3 was identified and the significant role in promoting electron transfer during HER process and accelerating HER kinetics by providing the hydrogen transfer pathway from H_xWO_3 onto Pt. However, The interaction between the locally inserted H species and Pt has not been carefully investigated. In our work, hydrogen intercalation not only fundamentally changes the interfacial charge transfer properties of WO_3 , but also functions as reactant to participate in HER process, resulting in the **coherent synergism between Ir and WO-H species**. Specifically, *operando* Raman measurements, selective poisoning and kinetic isotope effect experiments confirm the coherent synergism between Ir and lattice-hydrogen species of H_xWO_3 , that is, Volmer process is drastically boosted at Ir site to form Ir-H*, followed by spontaneous recombination of Ir-H* and neighboring revitalized WO-H* species to form H_2 molecular *via* interfacial Tafel step, as verified by theoretical calculations, thereby keeping the reaction at a high rate.

(2) *Nat. Commun.*, 2022, 13, 2024; *Nat. Commun.*, 2019, 10, 4876.

First of all, we would like to thank the reviewers for providing us with these two high-quality papers, which have made an indelible contribution to the regulation of the catalytic environment to improve the kinetics of catalytic reactions. Qiao et al. (*Nat. Commun.*, 2019, 10, 4876.) studied a series of Pt-based nanostructured electrocatalysts to reveal the unique water dissociation and proton reduction mechanism on nanomaterials. A unique acidic environment on the catalyst's surface was first identified under high $[\text{OH}^-]$ conditions by *in situ* Raman characterizations and electrochemical kinetic analysis (Figure R22). This work highlights that the **formation of local H_3O^+ is closely related to the bulk $[\text{OH}^-]$ of electrolyte**. In all alkaline electrolytes, water dissociation is always the most important step toward producing hydrogen source needed for HER process. Under high $[\text{OH}^-]$, large amounts of OH^- can be easily adsorbed to the catalyst, strongly promoting the water dissociation process. As water dissociation continues, more and more H ions are still bonded to nearby water

molecules but not to the catalyst's surface, which is being occupied by H_{upd} species. Thus a large amount of free H_3O^+ ions is generated within the double layer, resulting in an acid-like local environment. Under low $[OH^-]$, such as 0.01 M KOH and neutral buffer, the absence of local acidified environment is due to the sluggish water dissociation kinetics, indicating that achieving a local acidification environment under neutral/neutral conditions is quite challenging. To the best of our knowledge, **there are no relevant literature reports on the formation of a local acidification microenvironment under neutral/neutral conditions** to accelerate the reaction rate of neutral HER.

In Yan's work (*Nat. Commun.*, 2022, 13, 2024), corresponding characterizations and analysis confirm that the critical factors of the forming the local acid-like environment in Pt/MgO catalytic system include: the oxygen vacancy enriched MgO facilitates H_2O dissociation to generate H_3O^+ species; positively charged H_3O^+ migrates to negatively charged $Pt^{\delta-}$ and accumulates around $Pt^{\delta-}$ nanoparticles due to the electrostatic attraction, thus creating a local acidic environment in the alkaline medium (pH = 12, 13 and 14). It is obvious that the local degree of acidification in the Pt/MgO system significantly depends on the number of Pt-MgO interfacial sites and their exposure extent, as well as the strength of the physical electrostatic forces. This weaker physical field force obviously cannot be successfully transposed to neutral conditions where the ionic conductivity is much worse. In addition, the limited degree of acidification greatly obstacles its stable operation under high current conditions. In our work, WO_3 , which has a strong proton storage capacity and excellent proton conductivity, was chosen as the support to achieve the **maximum local acidification** nearby the metal sites under neutral conditions (Figure R23). Ir- H_xWO_3 exhibits highly competitive catalytic activity at high current density, reaching $\sim 360 \text{ mA cm}^{-2}$ at a potential of $-0.2 \text{ V}_{\text{RHE}}$ and operating stably for 40 h at an industrial-grade current density of 500 mA cm^{-2} .

Figure R22. (a) The in-situ Raman characterizations of Pt/C in 0.1 M and 0.01 M KOH; (b and c) the Tafel plots and E_a of catalysts in different $[\text{OH}^-]$ electrolyte; (d) Schematic illustration of the acid-like local environment mechanism on the nanostructured electrocatalysts (*Nat. Commun.*, 2019, 10, 4876).

Figure R23. Comparison of local acidic environments of $\text{Ir-H}_x\text{WO}_3$ and Pt/MgO catalyst.

Special changes are as follows:

Introduction section. In Page 3, Line 48:

“Therefore, selecting a suitable system to create a local acid-like environment through multiple physicochemical effects **to the maximum extent possible**, will provide an alternative way to promote the electrocatalytic performance and guide the higher efficiency electrocatalyst design in non-acidic electrolyte, especially in more challenging neutral media.”

In Page 3, Line 60-65

“Up to now, **it still encounters many problems and challenges, for example, 1) the degree of local acidification of H_xWO_3 in neutral media has not been accurately quantified; 2) the synergistic catalysis between local acidic species and co-catalysts has**

not been fully understood; 3) the enhanced activity cannot be simply attributed to a single optimized catalytic site, and the origin of the activity deserves further investigation.”

(2) The local acid-like microenvironment proposed by the authors lacks sufficient evidence. What is the pH of the local acid-like microenvironment? Is this local acid-like microenvironment stable during the HER processes?

Response:

Thank you very much for your professional question. We used rotating ring-disk electrode (RRDE) technique (*ChemElectroChem.*, 2019, 6, 4750; *Nat. Energy*, 2023, 8, 264) to quantitatively detect local pH on the H_xWO₃ and carbon support surfaces at different applied potentials in neutral PBS solution. According to convective-diffusion equation of Albery et al. (*J. Chem. Soc. Faraday Trans.* 1983, 79, 2583), the principle of the RRDE technology is based on the relationship between the pH values of the disk surface and the ring surface in solutions with arbitrary pH (Figure R24a). The calculation formula is as follows:

$$c_{rt, H^+} - c_{rt, OH^-} = N_D(c_{d, H^+} - c_{d, OH^-}) + (1 - N_D)(c_{\infty, H^+} - c_{\infty, OH^-}) \quad (1-1)$$

where c_{rt, H^+} and c_{d, H^+} are the concentrations of H⁺ on the RE and DE, respectively, c_{rt, OH^-} and c_{d, OH^-} are the concentrations of OH⁻ on the RE and DE, respectively, c_{∞, H^+} and c_{∞, OH^-} are the concentrations of H⁺ and OH⁻ in the bulk electrolyte, respectively, and $N_D = 0.37$ is the collection efficiency of the RE. Then, the pH dependence of the open circuit potential (E_{ocp}) in PBS solution was measured with Pt RE. Because in the H₂-saturated solution with inert electrolyte, the OCP of the Pt electrode would indicate the equilibrium potential (E) of $2H^+ + 2e^- \rightarrow H_2$, which varies with pH according to the Nernst equation:

$$E \text{ (V vs. SHE)} = \frac{-2.303RT}{F} \text{pH} \quad (1-2)$$

The fugacity of H_2 is assumed to be equal to unity and R , T and F are the gas constant, the absolute temperature and the Faraday constant. Hence, the relationship between the disk pH and the ring pH can be corroborated experimentally.

To confirm that the OCP of the Pt ring electrode accurately reflects the hydrogen equilibrium potential, the ring OCPs were measured with solutions of various pH values without any reactions at the disk electrode at room temperature ($\sim 20\text{ }^\circ\text{C}$). The time dependence of the ring OCP is shown in Figure R24b. The pH values shown in the figure are those in the bulk solution measured by a pH meter. Each OCP showed good stability. These OCPs were plotted against the pH values in Figure R24c, where the dotted line obeys the Nernst equation. The ring responses to the change in pH were in good agreement with the Nernst equation. Their linear regression had a slope of 52 mV/pH. The theoretical response according to Nernst equation at $25\text{ }^\circ\text{C}$ shows a slope of 59.2 mV/pH and an intercept of 0 mV. Our results were also in reasonable agreement with the results of Hessami et al (*AIChE J.*, 1993, 39, 149. the slope is 56.9 mV/pH).

Figure R24. (a) Schematic diagram for monitoring pH on the electrode surface (pH surface) using RRDE. During test, H^+ generated on the catalyst surface ($C_{H^+, \text{disk}}$) was carried away by radial electrolyte flow and detected by the ring electrode. (b) Time and (c) pH dependence of open circuit potential (E_{ocp}) for Pt-ring electrode. The measurement was performed in PBS solutions, and the pH of the PBS solutions was changed by adding H_2SO_4 or KOH .

Next, the local pH measurement was performed on the H_xWO_3 and carbon support in neutral pH solution during HER. A solution of H_2 -saturated 1.0 M PBS was used as an electrolyte. The catalyst loading is 0.51 mg cm^{-2} . The bulk pH of the solution is 7.03. **To obtain steady-state local pH value**, constant potential method was performed on the disk electrode, and OCP was measured simultaneously on the Pt ring electrode. In the constant potential method, each potential is maintained for 200 s ($E = 0.1, 0, -0.1, -0.2, -0.3, -0.4, -0.5, -0.6$ and $-0.7 V_{RHE}$) to obtain a steady-state current response (j). All measurements were carried out at rotation speed of 1600 rpm at room temperature. The relatively smooth i-t curves indicates that the electrode surface is in steady state at different potentials (Figure R25a and R25b). The measured local pH on cathode surfaces (H_xWO_3 and carbon) are shown in Figure R25c. For the H_xWO_3 support, the pH of the catalyst surface will be from 6.27 to 3.53 as the potential is reduced from 0.1 to $-0.4 V_{RHE}$, which undoubtedly confirms its local acid-like microenvironment. As the potential continues to shift negatively (-0.4 to $-0.7 V_{RHE}$), HER starts to occur and consumes the local hydrogen species, along with the increase in pH. When the potential reaches $-0.7 V_{RHE}$ ($j \sim 12 \text{ mA}$), the pH of the catalyst surface becomes 8.32, typical of an alkaline environment. In sharp contrast, the pH of carbon cathode surface is maintained at ~ 7 in range of $0.1 \sim -0.4 V_{RHE}$. As the potential continued to shift negatively, the pH of the carbon support surface gradually increased to 8.0 at $-0.7 V_{RHE}$ ($j \sim 5 \text{ mA}$). Astonishingly, when the bias ($-0.7 V_{RHE}$) is removed, the surface of H_xWO_3 and carbon cathodes turn back to steady acidic and neutral states, respectively (Figure R25d).

Figure R25. (a) i-t plots of H_xWO₃ support in different potentials; (b) i-t plots of carbon support in different potentials; (c) Measured pH values on H_xWO₃ and carbon cathode surfaces at different potentials, respectively; (d) The pH on H_xWO₃ and carbon cathode surfaces before and after removing bias (-0.7 V_{RHE}).

Special changes are as follows:

Methods Section. In Page 29-30, line 576-line 603:

“Measurement of pH on the catalyst surface

According to previous work⁴⁴, the pH values on the catalyst surfaces were measured by rotating ring-disk electrode (RRDE) technique. The potential of Pt ring electrode (RE) is sensitive to pH and can be used to monitor the variations in the pH on the disk electrode (DE) surface. A three-electrode cell was constructed of the RRDE, graphite rod and a saturated calomel electrode (SCE) as working, counter, and reference electrodes, respectively. Then, the pH dependence of the open circuit potential (E_{ocp}) in H₂-saturated 1.0 M PBS solution was measured with Pt RE (Supplementary Fig. 5). The OCP of the Pt electrode would indicate the equilibrium potential of $2H^+ + 2e^- \rightarrow H_2$, which varies with pH according to the Nernst equation:

$$E \text{ (V vs. SHE)} = \frac{-2.303RT}{F} \text{pH} \quad (1-1)$$

The fugacity of H₂ is assumed to be equal to unity and R , T and F are the gas constant, the absolute temperature and the Faraday constant.

For measuring the pH on the electrode surface, the investigated catalyst was loaded onto the disk electrode. The catalyst ink was prepared by ultrasonically dispersing catalyst powder (5 mg) in 5 wt% Nafion solution (20 μ L) and ethanol (480 μ L) mixed solution. 10 μ L of catalyst ink (10.0 mg mL⁻¹) was then transferred onto the disk electrode. The pH measurements on the catalyst surfaces were conducted in 1.0 M PBS solution with the working electrode rotating at a speed of 1600 rpm. Constant potential method was performed on the disk electrode ($E = 0.1, 0, -0.1, -0.2, -0.3, -0.4, -0.5, -0.6$ and -0.7 V_{RHE} for 200 s) to obtain a steady-state current response (j), and OCP was simultaneously measured on the Pt ring electrode. The pH value of the catalyst-loaded DE can be deduced from the pH value of the Pt RE by the following equation:

$$c_{rt,H^+} - c_{rt,OH^-} = N_D(c_{d,H^+} - c_{d,OH^-}) + (1 - N_D)(c_{\infty,H^+} - c_{\infty,OH^-}) \quad (1-2)$$

where c_{rt,H^+} and c_{d,H^+} are the concentrations of H⁺ on the RE and DE, respectively, c_{rt,OH^-} and c_{d,OH^-} are the concentrations of OH⁻ on the RE and DE, respectively, c_{∞,H^+} and c_{∞,OH^-} are the concentrations of H⁺ and OH⁻ in the bulk electrolyte, respectively, and $N_D = 0.37$ is the collection efficiency of the RE.”

Main text. In Page 6, line 110-line 124:

“Local pH measurements and catalytic behaviors of H_xWO₃

We used rotating ring-disk electrode (RRDE) technique^{19,44,45} to quantitatively detect local pH on the H_xWO₃ cathode surfaces at different applied potentials in neutral PBS solution (bulk pH is 7.03), along with classical carbon support for comparison (Supplementary Fig. 4-6, details see Methods). On the basis of RRDE detecting method, we found that the pH of the H_xWO₃ surface varies from 6.27 to 3.53 as potential decreases from 0.1 to -0.4 V_{RHE}, which undoubtedly confirms that H_xWO₃ acts as proton sponge to form a local acid-like microenvironment on the electrode surface (Fig. 2a,b). In sharp contrast, the pH of carbon cathode surface is maintained at ~7 in range

of 0.1 ~ -0.4 V_{RHE} (Fig. 2b). As the potential continued to shift negatively, the pH of H_xWO₃ and carbon support surfaces gradually increase and approach to 8.32 and 8.0 at -0.7 V_{RHE}, respectively, due to the consumption of the local hydrogen species. Astonishingly, when the bias (-0.7 V_{RHE}) is removed, the surface of H_xWO₃ and carbon cathodes turn back to steady acidic (pH=3.73) and neutral (7.12) states, respectively (Fig. 2c).”

The modified Figures. In Page 8, line 146-line152:

Fig. 2: Local pH measurements and catalytic behaviors of H_xWO₃. a, Schematic diagram of local acid-like microenvironment generation on H_xWO₃ cathode. b, Measured pH values on H_xWO₃ and carbon cathode surfaces at different potentials, respectively. c, The pH on H_xWO₃ and carbon cathode surfaces before and after removing bias (-0.7 V_{RHE}). d, HER polarization curves of H_xWO₃ grown on CFP and (e) corresponding Tafel plots. f, Free energy barriers of H-H coupling and H-H transfer in H_xWO₃.

In Page S6:

Supplementary Figure 4. Principles for detecting local pH. Schematic diagram for monitoring pH on the electrode surface (pH surface) using an RRDE technology.

In Page S7:

Supplementary Figure 5. The relationship between E_{ocp} of Pt ring electrode and pH. (a) Time and (b) pH dependence of open circuit potential (E_{ocp}) for Pt ring electrode. The measurement was performed in 1.0 M PBS solutions, and the pH of the PBS solutions was changed by adding H₂SO₄ or KOH.

In Page S8:

Supplementary Figure 6. The steady-state current of electrodes at different potentials. J - T

curves of H_xWO_3 (a) and carbon (b) support in different potentials.

The added references. In Page 37, line 755-line 759:

44. Yokoyama, Y., Miyazaki, K., Miyahara, Y., Fukutsuka, T. & Abe, T. In Situ Measurement of Local pH at Working Electrodes in Neutral pH Solutions by the Rotating Ring-Disk Electrode Technique. *ChemElectroChem* 6, 4750-4756 (2019).

45. Yokoyama, Y. et al. In situ local pH measurements with hydrated iridium oxide ring electrodes in neutral pH aqueous solutions. *Chem. Lett.* 49, 195-198 (2020).

(3) In the operando electrochemical Raman spectra, the WO-H in $Ir-H_xWO_3$ gradually disappeared in the voltage range, and the explanation given by the authors is that the surface WO-H species were depleted during the HER process. However, no obvious performance degradation was found in the long-term stability test, there exists a contradiction.

Response:

We sincerely thank the reviewer for the insightful comments and we are pleased to clarify this issue. A combination of *operando* electrochemical Raman spectra, selectively Li^+ and SCN^- poisoning experiments, kinetic isotope effect (KIE) experiments and density functional theory (DFT) calculations unambiguously unveiled that a novel neutral HER mechanism involved lattice WO-H species and metallic Ir site synergistic catalysis pathway in $Ir-H_xWO_3$. As mentioned by the reviewer, surface WO-H species were depleted during the HER process, resulting in the hydrogen-deficient state at the interface of $Ir-H_xWO_3$. This state can be eliminated by the hydrogen transfer on the surface of H_xWO_3 to replenish hydrogen, **thereby realizing the closed-loop as well as stable operation of the entire catalytic reaction**. As exhibited in Figure R26 and R27 and recently reported literature (*J. Phys. Chem. C*, 2014, 118, 1, 494; *J. Am. Chem. Soc.* 2021, 143, 24, 9236; *J. Mater. Chem. A*, 2019, 7, 23756; *Energy Mater. Sol. Cells* 1999, 56, 231; *J. Catal.* 1970, 17, 359), the demonstrated low barriers of H diffusion with water participation are consistent with the high proton-mobility in surface of WO_3 observed in experiment. That is to say, transport of H in H_xWO_3 cannot

be the rate determining step in the catalytic process proposed for Ir- H_xWO_3 system. To more visually demonstrate the continuous replenishment of hydrogen, chronopotentiometry test at 100 mA cm^{-2} was measured on Ir- H_xWO_3 catalyst for 10 s, and then observe the change of electrode before and after removing bias (Figure R28, Supplementary Movie 4 and 5). Counterintuitively, although the amount of bubbles on the surface of the Ir- H_xWO_3 electrode significantly reduced after stopping CP test, it continued to produce H_2 bubbles continuously for at least 1 minute. Using the commercial Pt/C system as a comparison, there was no continuous H_2 production on Pt/C catalyst surface when the bias was removed. In a word, The above discussion and experiments confirm that the **unique H compensation mechanism** of the H_xWO_3 support makes the local acidification microenvironment stable and effective.

Figure R26. Free energy barriers of H-H transfer in H_xWO_3 (data from Fig. 2f and Supplementary Figure 8).

Figure R27. Calculated energy profile of the H migration route $O_{1t} \rightarrow O_{1t'}$ with water participation. (*J. Phys. Chem. C* 2014, 118, 1, 494)

Figure R28. Photographs of Ir-H_xWO₃ and commercial Pt/C catalysts before and after chronopotentiometry test at 100 mA cm⁻² (The photographs are from Supplementary Movie 4 and 5).

Special changes are as follows:

Main text. In Page 21-22, line 424-line 426:

“The hydrogen-deficient state at the interface can be eliminated by the hydrogen transfer on the surface of H_xWO₃ to replenish hydrogen, thereby realizing the closed-loop of the entire catalytic reaction. Additionally, Supplementary Movie 4 and 5 visually demonstrate the continuous replenishment of hydrogen on Ir-H_xWO₃ surface after removing bias, along with Pt/C for comparison. Hence, the exceptional neutral HER electrocatalytic performance of Ir-H_xWO₃ resulted from the coherent synergism of Ir and in-situ formed lattice-hydrogen components.”

Supplementary Movie 4 and 5 are provided in supplementary video files.

(4) In the XRD pattern of Ir-H_xWO₃, Supplementary Figure 14a, why were the peaks of the carbon paper enhanced after the HER stability test?

Response:

We sincerely thank the reviewer for the insightful comments and we are pleased to clarify this issue. After HER stability testing, the XRD pattern of carbon paper is enhanced, which mainly stems from the fact that the surface loose H_xWO₃ nanorods break away from the whole catalyst skeleton under the continuous and violent impact of H₂ bubbles during the stability test, thus exposing the carbon paper substrate (Figure R29). The same phenomenon has also been reported in other works (*ACS Appl. Mater. Interfaces* 2018, 10, 14777; *Nanoscale*, 2021, 13, 8264; *J. Phys. Chem. C* 2016, 120,

16537; *J. Mater. Chem. A*, 2019,7, 775; *ACS Sustainable Chem. Eng.* 2018, 6, 11884; *Electrochim. Acta*, 2017, 241, 106. and Figure R30).

Figure R29. (a and b) The typical SEM images of Ir-H_xWO₃ before and after stability test.

Figure R30. The typical XRD patterns of catalysts before and after HER stability test in reported papers.

In order to more visually describe the structure changes of catalyst before and after stability tests, we slightly modified Supplementary Figure 20 in the revised

supplementary information (please see below).

Supplementary Figure 20. Structural analysis of catalysts after stability test. (a) The XRD patterns of Ir-H_xWO₃ before and after stability test. (b,c) The SEM and HRTEM images of Ir-H_xWO₃ after stability test, the inset in (b) giving the initial morphology of catalyst. (d) Line-scan profile of postmortem Ir-H_xWO₃ for Ir, W and O elements.

(5) Li⁺ and SCN⁻ poisoning experiments for H_xWO₃ and SCN⁻ poisoning experiments for Pt/C need to be carried to examine their HER mechanism proposed for H_xWO₃.

Response:

We thank the reviewer for the helpful suggestions. Based on the the reviewer's comments, Li⁺ and SCN⁻ poisoning experiments for H_xWO₃ and SCN⁻ poisoning experiments for Pt/C were supplied to support the catalytic mechanism proposed for Ir-H_xWO₃ systems. As we can see in Figure R31, after Li⁺ poisoning treatment, commercial 20 wt% Pt/C catalyst displays negligible activity decay. In contrast, the 20 wt% Pt/C catalyst undergoes SCN⁻ poisoning treatment and shows a drastic activity decrease due to the strong coordination capability between metal-centered catalytic sites and SCN⁻ species. For H_xWO₃ support, after Li⁺ and SCN⁻ poisoning treatments, it shows the opposite activity change trend with Pt/C catalyst (Figure R32), namely, the HER performance decrease of H_xWO₃ is charged by Li⁺ poisoning, rather than SCN⁻ poisoning. In summary, Li⁺ and SCN⁻ can selectively poison metal-centered catalytic

sites ($M + \text{SCN}^- \rightarrow M\text{-SCN}$) and surface acid species of H_xWO_3 ($\text{WO-H} + \text{Li}^+ \rightarrow \text{WO-Li} + \text{H}^+$). In Ir- H_xWO_3 catalytic system, the apparent activity decay was detected in both Li^+ and SCN^- poisoning experiments, unambiguously unveiling the coherent synergism between Ir and lattice-hydrogen species of H_xWO_3 (detailed analysis see Page 16-19).

Figure R31. (a) The LSV curves of commercial 20 wt% Pt/C before and after (a) Li^+ and (b) SCN^- poisoning in 1.0 M PBS solution, respectively.

Figure R32. (a) The LSV curves of H_xWO_3 before and after (a) Li^+ and (b) SCN^- poisoning in 1.0 M PBS solution, respectively.

Special changes are as follows:

Main text. In Page 18, line 348-line 350:

“The same process was performed on Pt/C and H_xWO_3 support in Supplementary Fig. 22 for comparison to confirm Li^+ selectively poisons the WO-H species and exclude its effect on the metal sites.”

In Page 18, line 354-line 356:

“For comparison, after SCN^- treatments for Pt/C and H_xWO_3 support in Supplementary Fig.23, an obvious activity reduction of Pt/C is detected and has no effect on H_xWO_3

support.”

The modified Figures. In Page S24:

Supplementary Figure 22. Effect of Li⁺ poisoning on the catalyst. The LSV plots of 20% Pt/C (a) and H_xWO₃ support (b) before and after Li⁺ poisoning treatments.

In Page S25:

Supplementary Figure 23. Effect of SCN⁻ poisoning on the catalyst. The LSV plots of 20% Pt/C (a) and H_xWO₃ support (b) before and after SCN⁻ poisoning treatments.

REVIEWER COMMENTS

Reviewer #1 (Remarks to the Author):

I recommend the publication of paper as authors have adequately modified it.

Reviewer #2 (Remarks to the Author):

I am satisfied with the revised manuscript and the authors' responds to reviewers' questions. I am happy to recommend the acceptance of this work as is.

Reviewer #3 (Remarks to the Author):

The authors' responses to the comments are reasonable and this manuscript can now be recommended for publication as is.

Reviewer #4 (Remarks to the Author):

Although the authors have improved their manuscript to some extent, I still feel that this article is not suitable for publication in Nature Communications for the following reasons:

1. The evidence that H in HxWO₃ involved in HER is insufficient, and whether the mechanism is correct requires further verification.
2. This group has ever published an article (Nature Communications (2022) 13:5382), very similar to this study in characterization methods, experiments, which significantly decrease the innovation of this study.
3. In their previous article, the authors suggested that the high HER activity of Ru/WO_{3-x} is originated from oxygen-deficient WO_{3-x} with a large proton storage capacity, they claim that the protons can be transferred to Ru NPs under cathodic potential, increasing the hydrogen coverage on the surface of Ru NPs in HER. In this manuscript, noble metal Ir was used instead of Ru, however, the authors suggest that the lattice hydrogen react at the interface, which is different from the mechanism they proposed for Ru/WO_{3-x}. The findings of this study are not at all consistent with those of Ru/WO_{3-x}, which suggests that further investigations are required.

Point-by-point response to the reviewer's comments

Comments by Reviewer's:

Reviewer #1 (Remarks to the Author): I recommend the publication of paper as authors have adequately modified it.

Response: Thank you very much for your positive recommendations.

Reviewer #2 (Remarks to the Author): I am satisfied with the revised manuscript and the authors' responds to reviewers' questions. I am happy to recommend the acceptance of this work as is.

Response: Thank you very much for your positive recommendations.

Reviewer #3 (Remarks to the Author): The authors' responses to the comments are reasonable and this manuscript can now be recommended for publication as is.

Response: Thank you very much for your positive recommendations.

Reviewer #4 (Remarks to the Author): Although the authors have improved their manuscript to some extent, I still feel that this article is not suitable for publication in Nature Communications for the following reasons:

We express our sincere gratitude to the reviewer for your careful review on our manuscript and the constructive comments, which really help to improve the quality of our manuscript.

Comment 1. The evidence that H in H_xWO_3 involved in HER is insufficient, and whether the mechanism is correct requires further verification.

Response: We sincerely thank the reviewer for the insightful comments and we are pleased to clarify this issue. Then, the detailed interpretations about participation of H_xWO_3 in HER process are presented from both the experimental and theoretical aspects, as below:

1) Experimental evidences

1.1 *Operando* electrochemical Raman measurements (Corresponding to the Figure 5 in the main text).

For H_xWO_3 comparison sample, the *Operando* Raman spectra results suggest that hydrogen insertion occurs in WO_3 in response to the cathodic voltage, where the H atoms are incorporated into the Brønsted acidic W^{5+} -OH groups (featured by emerging ~ 1580 and 2715 cm^{-1} peaks). However, the lack of H-H coupling sites and weak electrochemical activation prevents the surface hydride from being effective as H_2 molecular from the surface. After loading Ir nanoparticles, different variation trends of W-OH species are observed. The Raman signals of WO-H progressively weaken over whole potential range (OCP to -0.3 V_{RHE}), which further illustrates that the Ir- H_xWO_3 depleted surface W-OH species during HER process to alleviate deep-hydrogenation of H_xWO_3 supports. Typically, Ir NPs enhance the adsorption and dissociation of interfacial water molecules. Obviously, W-OH species and Ir NPs are simultaneously functioned as active sites to generate possible synergistic catalysis.

1.2 Selectively poisoning experiments (Corresponding to the Figure 6a,b in the main text and Supplementary Figure 22-24).

According to the Supplementary Figure 22 and 23, Li^+ and SCN^- ions can selectively poison W-OH and metal sites, respectively (*J. Am. Chem. Soc.* 2021, 143, 20133-20143; *Angew. Chem., Int. Ed.* 2020, 59, 8982-8990.). After Li^+ or SCN^- poisoning, the Ir- H_xWO_3 catalyst exhibits striking HER activity decay (Figure 6a), highlighting the importance roles of W-OH species and Ir metal in HER, which is consistent with the *Operando* Raman results. Then, the Tafel slopes of Ir- H_xWO_3 catalyst before and after Li^+ or SCN^- poisoning are compared to analyze the change of the catalytic reaction pathway. As shown in Fig. 6b, after WO-H replaced by WO-Li, the value of Tafel slope of catalyst increases from 28 to 89 mV dec^{-1} (similar to that of commercial Ir/C catalyst in neutral condition), manifesting that the rate-determining step (RDS) of the reaction changes from Tafel ($H_{ad} + H_{ad} \rightarrow * + H_2$) to Heyrovsky step ($H_{ad} + H_2O + e^- \rightarrow * + H_2 + OH^-$) and elucidating that the presence of lattice-hydrogen alters the catalytic mechanism. Furthermore, after poisoning the metal sites, the RDS transforms into

Volmer step (158 mV dec^{-1} , $^*\text{H}_2\text{O} + \text{e}^- \rightarrow ^*\text{H} + \text{OH}^-$), significantly unveiling that Ir metal site is response for water dissociation.

1.3 Kinetic isotope effect (KIE) experiments. (Corresponding to the Figure 6c,d in the main text and Supplementary Figure 25,26)

The participation of H_xWO_3 in HER process is further evidenced by KIE experiments. By switching the proton source from H_2O to D_2O , an obvious induction period (The KIE values from 2.57 to 5.16) is detected, which can be ascribed to surface active WO-H species are gradually replaced by WO-D, resulting in worse hydrogen transfer kinetics (Figure 6c,d). Assuming that W-OH is not involved in the production of H_2 , its KIE value should be at a stable value as Pt/C sample (Supplementary Figure 26). To further confirm the KIE effect, the pre-synthesized Ir- D_xWO_3 also manifests a similar induction period in 1.0 M PBS (H_2O) electrolyte, showing enhanced HER activity (Supplementary Figure 25)

2) Theoretical aspects. (Corresponding to the Figure 7 in the main text)

Then, two HER pathways (H-H coupling) are considered by DFT calculations (Figure 7). In contrast to the coupling of two hydrogen species at the metal site to generate H_2 , WO-H mediated interfacial Tafel step ($\text{WO-H}^* + \text{Ir-H}^* \rightarrow \text{W-O} + \text{Ir} + \text{H}_2$) experiences an exothermic process (-0.08 eV), which contributes to fast hydrogen production rate, as obtained experimentally. More importantly, The hydrogen-deficient state at the interface can be eliminated by the hydrogen transfer on the surface of H_xWO_3 to replenish hydrogen, thereby realizing the closed-loop of the entire catalytic reaction.

Based on the conclusions obtained from the above experimental and theoretical studies, we propose a possible interfacial hydrogen-evolution pathway mediated by lattice hydrogen of neutral HER catalyzed by hybridized Ir- H_xWO_3 . The Ir metal component in the hybridized Ir- H_xWO_3 possesses superior electrocatalytic activity toward the Volmer process and is expediently utilized to strongly adsorb H_2O , effectively catalyze the dissociation of H_2O^* to generate interfacial Ir- $^*\text{H}$. Thermodynamically favorable

Tafel process is advantageously utilized to efficiently combine the interfacial Ir-H* and neighboring reactive WO-H* into H₂. Unlike the well-known lattice-oxygen-mediated oxygen evolution reaction, the neutral HER mechanism mediated by lattice-hydrogen has been rarely reported. Typically, Patrik Schmuki et al. reported that (*J. Mater. Chem. A*, 2020, 8, 22773-22790) hydrogen-rich TiO₂ surface not only stabilizes the deposited Ir and weakens its H binding strength to a moderate intensity, but also actively takes part in the HER mechanism by refreshing the Ir catalytic sites near the Ir|H-TiO₂ interface, thus substantially promoting H₂ generation in acidic media (Figure R1). However, there still lack of solid experimental evidences to prove interfacial H₂ evolution process. Herein, a combination of *operando* Raman measurements, selective poisoning and kinetic isotope effect experiments confirm the coherent synergism between Ir and lattice-hydrogen species of H_xWO₃. **We believe our work can open up new perspectives for HER mechanism research.**

Figure R1. Schematic representation of the reaction pathway at the Ir|H-TiO₂ interface. (*J. Mater. Chem. A*, 2020, 8, 22773-22790).

Comments 2 and 3. This group has ever published an article (Nature Communications (2022) 13:5382), very similar to this study in characterization methods, experiments, which significantly decrease the innovation of this study. In their previous article, the

authors suggested that the high HER activity of Ru/WO_{3-x} is originated from oxygen-deficient WO_{3-x} with a large proton storage capacity, they claim that the protons can be transferred to Ru NPs under cathodic potential, increasing the hydrogen coverage on the surface of Ru NPs in HER. In this manuscript, noble metal Ir was used instead of Ru, however, the authors suggest that the lattice hydrogen react at the interface, which is different from the mechanism they proposed for Ru/WO_{3-x}. The findings of this study are not at all consistent with those of Ru/WO_{3-x}, which suggests that further investigations are required.

Response: We sincerely thank the reviewer for the insightful comments and we are pleased to clarify this issue. Firstly, it is worth stating that the surface/interface chemistry of heterogeneous catalysts is highly sophisticated, and changes in components can significantly alter the catalytic activity and mechanism (*Chem. Soc. Rev.*, 2014, 43, 7870-7886; *Chem. Rev.* 2018, 118, 4981-5079.). As the reviewer mentioned that, our previous work (*Nat. Commun.*, 2022, 13, 5382) developed an effective strategy to significantly increase the H coverage on the catalyst during HER in neutral environment. The oxygen-deficient WO_{3-x} possesses a large capacity for storing protons, which can be transferred to the surface of Ru NPs under cathodic potential. This hydrogen spillover from WO_{3-x} to Ru changes the RDS of HER on Ru in neutral medium from water dissociation to hydrogen recombination (Heyrovsky step: $H_{ad} + H_2O + e^- \rightarrow * + H_2 + OH^-$), which greatly improves the HER kinetics (see Figure R2). This interesting experimental phenomenon and conclusion motivated us to further utilize localized hydrogen species of H_xWO₃ support to realize thermodynamically more favorable Volmer-Tafel (~ 30 mV/dec) steps (Ru-WO_{3-x} follows Volmer-Heyrovsky steps, ~ 40 mV/dec), thus obtaining acid-like HER activity and kinetics in challenging neutral media. Additionally, the interaction mechanism between metal and localized acidic species is still unclear, which is worthy of further exploration.

Figure R2. (a) LSV curve and the corresponding Tafel plot for commercial Ru/C (5.0 wt.%) loaded on CP; (b) The corresponding Tafel plots of commercial Ru/C, WO_{3-x}/CP and Ru-WO_{3-x}/CP; (c) Schematic diagram showing how hydrogen spillover from WO_{3-x} to Ru enhances HER in neutral environment. (*Nat. Commun.*, 2022, 13, 5382).

In order to achieve these goals, the composition as well as the structure of the catalyst should be designed rationally. Iridium (Ir), with its weak hydrogen binding energy, was first selected for the chemisorption of hydrogen and exhibits relatively balanced hydrogen adsorption/desorption capacity, comparable to Pt(100) (*J. Chem. Phys.* 1987, 87, 3104). Thus, similar to Pt, Ir metals that adsorb hydrogen neither strongly nor weakly occupy the apex of the classical HER volcano curve proposed by J. K. Nørskov (*J. Electrochem. Soc.*, 2005, 152, J23). While the metals that adsorb hydrogen strongly (Ru and 3d elements) are positioned on the descending part of the volcano plot (*Angew. Chem. Int. Ed.* 2012, 124, 12663). Therefore, the stronger H adsorption capacity of Ru makes the reversible hydrogen spillover phenomenon in Ru-WO_{3-x} more reasonable, while the H species enriched on Ru cannot be directly coupled to generate H₂ molecular. In comparison, Pt and Ir metals are more favorable for the expedited hydrogen desorption kinetics (Tafel step). Moreover, the adsorption/activation of interfacial water molecular is also a crucial step in HER process under neutral condition (*Adv. Energy Mater.* 2023, 13, 2203164; *Energy*

Environ. Sci., 2020, 13, 3185). Compared with Pt, higher oxygenophilic properties of Ir confer a lower water dissociation energy barrier, thus accelerating Volmer step. In summary, the moderate hydrogen adsorption strength and strong oxygenophilic features of Ir make it a promising candidate for neutral HER catalysts and can be adopted as an ideal model to study catalytic mechanism during the HER process. In this work, a novel Ir-H_xWO₃ catalyst is readily synthesized. Elaborated Ir-H_xWO₃ demonstrates **acid-like activity and kinetics** with ultralow overpotential of 20 mV at 10 mA cm⁻² and low Tafel slope of 28 mV dec⁻¹, which follows thermodynamically favorable Volmer-Tafel steps. To the best of our knowledge, this remarkable performance (overpotential and Tafel slope) is leading among the current neutral HER catalysts (Supplementary Table 2). A combination of physicochemical characterization and theoretical simulation confirm the new interfacial hydrogen-evolution pathway mediated by lattice-hydrogen of H_xWO₃ (detail analysis see the responses to question 1) and reveal the interaction mechanism between metal and localized acidic species.

To further eliminate the concerns of reviewers, the temperature programmed desorption (TPD) of H₂ was conducted to gain further insight into H₂ desorption sites in catalysts. As shown in Figure R3, The H₂-TPD profile of commercial Ir/C catalyst (microstructure see Figure R4) exhibits a single desorption peak centered at 127 °C, which is ascribed to the desorption of atomic H over the metallic Ir surface. As we expected in Ir-H_xWO₃ sample, in addition to the observation of H₂ desorption site located on the metal surface, the unique dehydrogenation signal over interface of Ir metal and H_xWO₃ support (155 °C) is also detected, which is consistent with previous results in the literature (*J. Catal.*, 2008, 260, 141-149; *Appl. Catal. A-Gen.*, 2017, 537, 59-65; *Top. Catal.*, 2023, 66, 205-222.). Moreover, by comparing the amount of dehydrogenation at different sites, we found that H₂ desorption at the interface site is dominant.

Figure R3 and revised Supplementary Figure 28. Analysis of H₂-TPD data. The H₂-TPD patterns of Ir-H_xWO₃, commercial Ir/C and H_xWO₃ samples.

Figure R4. The TEM image of commercial Ir/C catalyst.

Accordingly, we added Figure R3 as Supplementary Figure 28 in the revised supplementary information (in **Page S30**). we have added the statements in the revised manuscript and supplementary information, the specific as follows:

Main text. In Page 21, line 413-line 415:

“This unique and dominant interfacial dehydrogenation site is further corroborated by H₂-TPD experiments in Supplementary Figure 28.”

Method section. In Page 28, line 547-line 551:

“In the experiments to investigate hydrogen desorption sites, after Ar pretreatment, H₂ adsorption was carried in H₂ gas flow (30 ml min⁻¹) for 1 h, followed by purging with Ar gas (30 ml min⁻¹) for 1 h to remove the physically adsorbed H₂ on the surface. Finally,

a programmed temperature desorption test under Ar gas flow was performed.”

Supplementary information.

In Page 30: “Supplement: The H₂-TPD profile of commercial Ir/C catalyst exhibits a single desorption peak centered at 127 °C, which is ascribed to the desorption of atomic H over the metallic Ir surface. As we expected in Ir-H_xWO₃ sample, in addition to the observation of H₂ desorption site located on the metal surface, the unique dehydrogenation signal over interface of Ir metal and H_xWO₃ support (155 °C) is also detected, which is consistent with previous results in the literature¹⁻³. Moreover, by comparing the amount of dehydrogenation at different sites, we found that H₂ desorption at the interface site is dominant.”

In Page 37: 4. Supplementary References

1. Panagiotopoulou, P. & Kondarides, D. I. Effects of alkali additives on the physicochemical characteristics and chemisorptive properties of Pt/TiO₂ catalysts. *J. Catal.* **260**, 141-149 (2008).
2. Li, W. et al. Skeletal isomerization of n-pentane: a comparative study on catalytic properties of Pt/WO_x-ZrO₂ and Pt/ZSM-22. *Appl. Catal. A-Gen* **537**, 59-65 (2017).
3. Dolsirittigul, N. et al. Structure-Activity Relationships of Pt-WO_x/Al₂O₃ Prepared with Different W Contents and Pretreatment Conditions for Glycerol Conversion to 1, 3-Propanediol. *Top. Catal.* **66**, 205-222 (2023).